# Landscape of Interactions between Stromal and Myeloid Cells in Ileal Crohn’s Disease; Indications of an Important Role for Fibroblast-Derived CCL-2

**DOI:** 10.3390/biomedicines12081674

**Published:** 2024-07-26

**Authors:** Nikolas Dovrolis, Vassilis Valatas, Ioannis Drygiannakis, Eirini Filidou, Michail Spathakis, Leonidas Kandilogiannakis, Gesthimani Tarapatzi, Konstantinos Arvanitidis, Giorgos Bamias, Stergios Vradelis, Vangelis G. Manolopoulos, Vasilis Paspaliaris, George Kolios

**Affiliations:** 1Laboratory of Pharmacology, Department of Medicine, Democritus University of Thrace, 68100 Alexandroupolis, Greece; efilidou@med.duth.gr (E.F.); mspathak@med.duth.gr (M.S.); lkandilo@med.duth.gr (L.K.); gtarapat@med.duth.gr (G.T.); karvanit@med.duth.gr (K.A.); emanolop@med.duth.gr (V.G.M.); gkolios@med.duth.gr (G.K.); 2Individualised Medicine & Pharmacological Research Solutions Center (IMPReS), 68100 Alexandroupolis, Greece; 3Gastroenterology and Hepatology Research Laboratory, Medical School, University of Crete, 71003 Heraklion, Greece; idrygiannakis@gmail.com; 4GI Unit, 3 Department of Internal Medicine, Sotiria Hospital, National and Kapodistrian University of Athens, 11527 Athens, Greece; gbamias@gmail.com; 5Second Department of Internal Medicine, University Hospital of Alexandroupolis, Democritus University of Thrace, 68100 Alexandroupolis, Greece; svradeli@med.duth.gr; 6Tithon Biotech Inc., San Diego, CA 92127, USA; bpaspa@tithonbiotech.com

**Keywords:** cell–cell communication, Crohn’s disease (CD), pro-inflammatory differentiation, stromal cells, single-cell transcriptomics

## Abstract

Background and aims: Monocyte recruitment in the lamina propria and inflammatory phenotype driven by the mucosal microenvironment is critical for the pathogenesis of inflammatory bowel disease. However, the stimuli responsible remain largely unknown. Recent works have focused on stromal cells, the main steady-state cellular component in tissue, as they produce pro-inflammatory chemokines that contribute to the treatment-resistant nature of IBD. Methods: We studied the regulation of these processes by examining the communication patterns between stromal and myeloid cells in ileal Crohn’s disease (CD) using a complete single-cell whole tissue sequencing analysis pipeline and in vitro experimentation in mesenchymal cells. Results: We report expansion of S4 stromal cells and monocyte-like inflammatory macrophages in the inflamed mucosa and describe interactions that may establish sustained local inflammation. These include expression of CCL2 by S1 fibroblasts to recruit and retain monocytes and macrophages in the mucosa, where they receive signals for proliferation, survival, and differentiation to inflammatory macrophages from S4 stromal cells through molecules such as MIF, IFNγ, and FN1. The overexpression of CCL2 in ileal CD and its stromal origin was further demonstrated in vitro by cultured mesenchymal cells and intestinal organoids in the context of an inflammatory milieu. Conclusions: Our findings outline an extensive cross-talk between stromal and myeloid cells, which may contribute to the onset and progression of inflammation in ileal Crohn’s disease. Understanding the mechanisms underlying monocyte recruitment and polarization, as well as the role of stromal cells in sustaining inflammation, can provide new avenues for developing targeted therapies to treat IBD.

## 1. Introduction

Inflammatory bowel disease (IBD), a chronic autoinflammatory disorder of the gastrointestinal tract includes two main phenotypes: Crohn’s disease (CD) and ulcerative colitis. Traditionally, research and treatment in IBD has been focused on classic immune cells, particularly T cells, which are known to play a critical role in triggering and perpetuating autoimmune mucosal inflammation [1]. Over the last decade, accumulating evidence has suggested that stromal cell activation plays a crucial role not only in intestinal mucosal homeostasis but also in the pathogenetic mechanisms of treatment-resistant IBD [2]. Stromal cells are a heterogeneous group of non-classic immune cells found in all tissues, including the intestine, where fibroblasts and myofibroblasts are the predominant cell types [3]. In addition to structural support, they participate in immune responses by producing cytokines, chemokines, trophic, and other signaling molecules depending on their versatile functional specialization [4,5]. In IBD, activated stromal cells interact with their *locale* through various cell-to-cell communication pathways [6,7].

Although the intestine becomes seeded during development by embryonic precursors of macrophages, only a fraction of those is maintained to adulthood [8]. A substantial fraction of intestinal macrophages derive from circulating adult monocytes and are constantly replenished during adulthood depending on the niches they reside in [9]. Specifically in the lamina propria (LP), the “physiological inflammation” occurring from the constant exposure to luminal antigens results in a shortened macrophage lifespan. Therefore, continuous recruitment of monocytes to the intestinal mucosa is necessary during homeostasis that is further intensified in the context of intestinal inflammation such as IBD [10,11]. Monocyte recruitment and polarization towards inflammatory phenotypes is an orchestrating event in IBD that depends on the microenvironment of the lamina propria (LP) [12]. The accountable signals and their cellular origin are now under intense investigation [13]. Chemokine gradients by *CCL2*, *CCL5*, and *CX3CL1* have been implicated in the pathogenetic mechanisms of intestinal inflammation [14]. In addition to chemokine-mediated recruitment of monocytes, pro-inflammatory cytokines of the LP microenvironment have been suggested to exert an intestine-specific reprogramming of transcription in monocytes [12].

There is evidence that stromal cells produce soluble mediators that could potentially recruit and polarize monocytes [15]. Stadler et al. have demonstrated that colonic fibroblasts are the major cellular component in the gut mucosa recruiting and polarizing infiltrating monocytes to well-defined phenotypes of macrophages via shedding CD163 and producing CCL2 [16]. Numerous studies have failed to reveal a single factor responsible for this recruitment pattern, suggesting a mixture of mediators involved. Reports on stromal–myeloid cell communication pathways are lacking, and this may be relevant to the uprising of the resistance of IBD to single-cytokine neutralization therapeutic approaches. Thus, alternative proinflammatory pathways should be thoroughly investigated and stromal cells should be dissected into subtypes to shed light on novel therapeutic approaches [17].

In recent times, remarkable advancements in technology have opened up new avenues for in-depth exploration of complex phenomena. Single-cell RNA sequencing (scRNA-seq) is such a powerful addition to the ever-expanding scientific arsenal, enabling research to delve into the transcriptional landscape of specific cell populations in unprecedented detail. This groundbreaking technique has revolutionized our ability to study complicated processes, including inflammation, by simultaneously enlightening intra- and extra-cellular mechanisms across various conditions [18], including IBD [19].

The phenotypic hallmark of Crohn’s disease is the discontinuous involvement of the intestinal mucosa by chronic inflammation that results in areas of inflammation intercepted or followed by areas of non-inflamed normal-appearing tissue. Although inflamed and non-inflamed loci share the same genetic background and microflora, the non-inflamed areas are dominated by homeostatic mechanisms that promote immunoregulation, whereas in the inflamed areas events of immune dysregulation prevail. We believe that the delineation of cell–cell interactions in inflamed and non-inflamed areas in Crohn’s disease may represent an important tool to unravel those pathways that preferentially support CD development.

We hypothesized that stromal cells may play a prominent role in the signaling that drives monocyte recruitment and differentiation to inflammatory macrophages and employed a multifaceted approach to investigate. We introduce a comprehensive workflow that capitalizes on scRNA-seq to examine the dominant communication pathways between stromal and myeloid cells. The findings presented herein shed light on the functional characteristics of these cell populations during both homeostasis and inflammation, elucidating their intricate interactions through molecular crosstalk. Moreover, we picked up the most significant in silico findings and further explored them with our wet lab experiments. The insights gained have the potential to pave the way for novel therapeutic strategies targeting alternative molecular pathways.

## 2. Materials and Methods

### 2.1. Single-Cell RNA Sequencing Data Acquisition, Preprocessing, and Cell Clustering

Single-cell RNA sequencing (scRNA-seq) data (GSE134809) from paired (inflamed and non-inflamed mucosa) ileal biopsies from 11 CD patients with diseases confined to the small intestine were retrieved from the Gene Expression Omnibus [20], originating from the work of Martin et al. [21]. The term “inflamed” indicates that cells are obtained from areas with inflamed intestinal mucosa affected by Crohn’s disease, whereas the term “non-inflamed” indicates cellular populations obtained from adjacent intestinal areas of normal-looking mucosa of the CD patients that were not affected by inflammation. The data were imported into R v.4.2.1 [22] using the Read10X function from the Seurat package v.4.3.0 [23] with the latter also used to process the data and perform initial analyses. We decided on a non-integration approach which allowed us to explore inflamed and non-inflamed tissues separately on a biological basis. Our aim was to study cell behavior in the presence and absence of inflammation, independently. Integration approaches primarily try to align datasets with similar cell types or states so that they can be analyzed together as it is noted in the Seurat tutorials. We wanted to make sure that populations that existed in inflamed and non-inflamed tissues were not erroneously equated or discarded. So, for each separate dataset, the following steps were taken: Raw gene expression data were first filtered to remove low-quality (dead and doublet) cells based on the total number of genes detected per cell and the percentage of mitochondrial genes expressed. Cell types outside the scope of this work were also filtered out using the gene expression of typical erythrocyte (all hemoglobin genes matching the pattern “^HB [^(P)]” and *GYPA^+^*), platelet (*PECAM1^+^*, *PF4^+^*), epithelial (*EPCAM^+^*), T- (*CD3D^+^*, *CD27^+^*), B- (*CD19*^+^, *CD27^+^*), neural and natural killer (*NCAM1^+^)*, and hematopoietic progenitor (*FLT3^+^*) cell markers. Subsequently, the remaining cells were normalized with sctransform [24] and scaled to remove unwanted technical variation. Principal component analysis (PCA) was performed on the highly variable genes to identify the principal components (PCs) capturing the most significant sources of variation. Dimensionality was reduced with UMAP. The top PCs were used to cluster cells using Seurat’s graph-based clustering.

### 2.2. Cluster Refinement and Cell Type Annotation

Leftover, out-of-scope cell populations from initial filtering and clustering were removed after rough annotation using SingleR v.1.10.0 [25] and Celldex v.1.6.0 [25] with the BLUEPRINT/ENCODE dataset (retrieved on 25 September 2022). Only stromal and myeloid populations were selected for further analysis. Each individual object was reclustered to ensure that there were no residual populations and each cluster was annotated manually with the markers provided by Elmentaite et al. [26]. To characterize stromal clusters, the expression of genes such as *ADAM28*, *ADAMDEC1*, *CXCL14*, *CCL11*, *CFD*, *PDGFRA*, *BMP4*, *F3*, *NPY*, *CH25H*, *C7*, *KCNN3*, *C3*, *MMP1*, *PDPN*, *ZEB2*, *ACTA2*, *TAGLN*, *MYH11*, and *RSPO*2 was analyzed. For the myeloid clusters, the expression of genes including *S100A4*, *S100A6*, *S100A8*, *S100A9*, *LYZ*, *FCN1*, *TSPO*, *IL1B*, *ELANE*, *MPO*, *PTRN3*, *AZU1*, *RETN*, *CLEC9A*, *XCR1*, *IRF8*, *CLEC10A*, *LAMP3*, *CCR7*, *FSCN1*, *CLEC4C*, *IL3RA*, *JCHAIN*, *MZB1*, *IRF7*, *CD163*, *C1QB*, *C1QC*, *HLA-DRB1*, *CCL24*, *IL10RA*, *LYVE1*, *SPP1*, *MMP9*, *TPSB2*, *TPSAB1*, *GATA2*, *CPA3*, *HPGDS*, *CLC*, *LCN2*, *GP9*, and *PF4* was examined.

### 2.3. Differential Gene Expression within Populations

Even though cell population characterization was conducted using specific markers, as previously described, we wanted to further explore transcriptomic findings based on differential expression which uniquely discerns specific subpopulations. To achieve that, we specified four distinct groups within which we performed differential expression exclusively among their members: inflamed myeloid, inflamed stromal, non-inflamed myeloid, and non-inflamed stromal. In essence, this categorization aimed to capture the nuanced variations in gene expression patterns among closely related cell types in each tissue separately to delineate the specific role of each subset during inflammation or homeostasis.

To systematically uncover these differentially expressed genes, we employed the FindMarkers function of Seurat employing the Wilcoxon Rank Sum test. This function facilitates the identification of genes that exhibit significant expression differences between the specified subsets on a one versus one basis considering all possible group combinations. So, for example, we tested all non-inflamed stromal cells with each other. Our focus extended to the top ten positive-expression (parameter only.pos = TRUE) differentially expressed genes within each subset, based on their log_2_(Fold Change), as these were deemed crucial in delineating the distinct genetic characteristics associated with inflammation status within stromal and myeloid cell populations. All analyses used a false discovery rate (FDR) of <0.05 as a cut-off.

### 2.4. Intercellular Communication Analysis with CellChat

To analyze intercellular communication patterns, the CellChat package v.1.1.3 [27] was employed on the four cell groupings previously described. CellChat integrates ligand–receptor interactions and intracellular signaling pathways to infer outgoing and incoming signaling events between cell groups. Initially, CellChat utilizes the scRNA-seq data to predict the ligands expressed by sender cells and the receptors expressed by receiver cells. It then infers potential communication events by matching ligand–receptor pairs. Additionally, CellChat incorporates information on cofactors and intracellular signaling pathways to further refine the signaling predictions. The pipeline incorporates numerous features, such as identifying communication modules, linking cell groups to signaling pathways, and differentiating between outgoing and incoming signals. These capabilities were utilized to generate diagrams that explore the “number of interactions” between cell populations, their “interaction weights/strength” (aggregateNet and netVisual_circle functions), as well as a condensed representation of the pathways that facilitate said interactions (netAnalysis_computeCentrality and netAnalysis_signalingRole_heatmap functions). In the “number of interactions” diagram, the width of the chords corresponds to the number of interactions between cell clusters. In the “interaction weights/strength” diagram, the thickness of the chords conveys the significance of the interactions based on the probability of an interaction occurring. This probability is based on the law of mass action which models the interaction between cell groups by considering the average expression values of a ligand, a receptor, and their cofactors, quantifying the strength and likelihood of binding or signaling events. Finally, the heatmaps produced by the third approach illustrate the outgoing and incoming signaling patterns instituted by specific molecular pathways. Outgoing signaling refers to the signaling events initiated by a particular cell type towards other cell types, while incoming signaling represents the signals received by a specific cell type from other cell types based on specific pathways which are represented on the Y-axis. Each pathway is characterized by one or several ligand–receptor pairs. The probability of interaction is visualized by grading the tile opacity, and the height of a separate set of bars depicts the number of pathways each cell type sends or receives. Signals sent by a specific cell type in the outgoing panel are received by the cell types of the incoming panel in the same row (pathway).

### 2.5. Network Inference and Analysis with the NicheNet Algorithm

To further explore the intercellular communication network, we employed the nichenetr package v.1.1.1 [28]. The NicheNet algorithm leverages the scRNA-seq data and various complementary data sources, including ligand–receptor interactions, signaling pathways, and gene regulatory information to construct integrated networks. Within these networks, regulatory potential scores are calculated to assess the strength of ligand–target correlations. By employing this prior knowledge, NicheNet is capable of deducing active ligand–target links between interacting cells. The prioritization of ligands in NicheNet is based on their ability to accurately predict observed gene expression changes in the recipient cell (activity). Consequently, NicheNet identifies potential targets that are likely to be regulated by prioritized ligands on the grounds of high regulatory potential. Figure 1 illustrates the whole pipeline up to this step as a flowchart.

### 2.6. CCL2 RNA Sequencing

Raw RNA sequencing data (available at: https://www.ebi.ac.uk/ena/browser/view/PRJEB56386, accessed on 24 July 2024) from our previous work [29] were reanalyzed to identify the differential expression of *CCL2* in affected terminal ilea of CD patients versus healthy individuals. In brief, of all sequences, ileal samples were picked up, and aligned with Salmon v.1.6.0 on the GRCh38 Human Transcriptome reference database. The output was imported into R using tximport v1.20.0 [30] as a DESEQ2 v.1.32.0 [31] object on which differential expression analysis was performed.

### 2.7. Stromal Cell Isolation and Culture

Stromal cells were isolated from ileal biopsies of healthy individuals and patients with Crohn’s Disease, as previously described [32]. Briefly, tissues were washed ×3 in HBSS with Ca^++^/Mg^++^ supplemented with penicillin, streptomycin (Pen-Strep), amphotericin B (A), and gentamycin (G) and for another 3 times in the same medium without Ca^++^/Mg^++^ (Biosera, Nuaille, France) and then de-epithelialized using dithiothreitol (DTT) 1 mM and, subsequently, ethylene-diaminetetraacetic acid (EDTA) 1 mM (Sigma-Aldrich, Darmstadt, Germany). Tissues were set to culture in RPMI 1640 (PAN Biotech, Aidenbach, Germany), supplemented with 10% *v*/*v* Fetal Bovine Serum (FBS; Sigma-Aldrich, Darmstadt, Germany), and Pen-Strep/A in 5% CO_2_ at 37 °C until numerous myofibroblast colonies appeared on the flask culture surface. All experiments were performed with stromal cells at passages 2–5. Table 1 shows the CD patients’ characteristics.

### 2.8. Tissue Culture

Tissue culture supernatants from intestinal biopsies from normally appearing mucosa of healthy individuals and inflamed mucosa of patients with Crohn’s Disease were collected as previously described [33]. Briefly, tissues were collected in HBSS (Biosera, Nuaille, France) supplemented with Pen-Strep/A/G, washed six times, and cultured in RPMI 1640 (PAN Biotech, Aidenbach, Germany) supplemented with 10% *v*/*v* FBS and P/S/A/G for 30 h (h) in 5% CO_2_ at 37 °C. Then tissue samples and supernatants were collected and stored at −80 °C. Table 2 shows the CD patients’ characteristics.

### 2.9. Differentiation of the H1 Embryonic Cell Line to Human Intestinal Organoids

HIOs were developed from the H1 embryonic stem cell line (Wicell, Madison, WI, USA) using the STEMdiff™ Intestinal Organoid Kit (StemCell Technologies, Vancouver, BC, Canada), as previously described [34]. In short, H1 were primarily cultured in 6-well plates coated with Matrigel (Matrigel™; Corning, NY, USA) and supplemented with mTeSR™1 medium (StemCell Technologies, Vancouver, BC, Canada), and when the appropriate confluency was reached, the differentiation protocol began. First, H1 were differentiated into definitive endoderm (DE), by applying daily the endoderm basal medium which contained Activin A, and after 3 days when DE was formed, Activin A was replaced with Wnt3A and fibroblast growth factor 4 (FGF4). DE was maintained in this medium for 5–6 days until Mid-/Hindgut (MH) spheroids were starting to form in the supernatant. When an appropriate number of MH spheroids were formed, spheroids were collected, seeded into Matrigel (Corning, NY, USA) domes, and cultured in Intestinal Organoid Basal (IOB) medium supplemented with epidermal growth factor (EGF) and Noggin, until HIOs were formed. HIOs were cultured and passaged every 10 days at a 1:3 ratio.

### 2.10. Primary Human Colonoids

Colonoids were formed using the IntestiCult™-SF Organoid Growth Medium (StemCell Technologies, Vancouver, BC, Canada) according to the manufacturer’s instructions. Briefly, colonic endoscopic biopsies, from three healthy individuals with normal endoscopy, were collected in HBSS (Biosera, Nuaille, France), supplemented with Penicillin/Streptomycin (Biosera, Nuaille, France), Amphotericin B (Biosera, Nuaille, France), and Gentamycin (Biosera, Nuaille, France) and washed two times with ice-cold Dulbecco’s Phosphate-Buffered Saline (DPBS; Biosera, Nuaille, France). Using sterile scissors, biopsies were thoroughly minced into smaller pieces, and then washed and centrifuged in ice-cold DPBS at 300× *g* for 5 min. Supernatant was discarded and ice-cold Gentle Cell Dissociation Reagent (GCDR; StemCell Technologies, Vancouver, BC, Canada) was added, and tissue pieces were incubated for 30 min on a rocking platform. Next, tissue pieces were centrifuged at 300× *g* for 5 min, the supernatant was discarded, and were DMEM supplemented with 0.1% Bovine Serum Albumin (BSA; PAN Biotech, Aidenbach, Germany) was added. Tissue pieces were vigorously pipetted up and down several times, then passed through a 70 μm strainer (SPL Life Sciences, Pocheon-si, Republic of Korea), and finally, colonic crypts were isolated and seeded into Matrigel (Corning, NY, USA) domes. Colonic crypts were cultured in IntestiCult™-SF Organoid Growth Medium (StemCell Technologies, Vancouver, BC, Canada) until colonoids were formed after 10 days. Colonoids were cultured and passaged every 10 days at a 1:4 ratio.

### 2.11. Cytokine Treatments

Stromal cells from healthy individuals and CD patients, HIOs, and colonoids were cultured in FBS- and growth factor-free medium for 24 h, and then were either left untreated or stimulated with TNF-α 50 ng/mL and IL-1α 5 ng/mL (R&D Systems, Minneapolis, MN, USA) for 6 or 24 h. At 6 h, cells were lysed with the LB1 buffer of the NucleoSpin^®^ RNA Plus XS kit (Macherey-Nagel, Düren, Germany) and at 24 h, supernatants were collected for Enzyme Linked Immunosorbent Assay (ELISA). Both were stored at −80 °C until further processing.

### 2.12. RNA Extraction

Total RNA from stromal cells, HIOs, and colonoids was extracted using the NucleoSpin^®^ RNA Plus XS kit (Macherey-Nagel, Düren, Germany), according to the manufacturer’s instructions. Briefly, an equal volume of LB2 was added to samples lysed in LB1, and samples were centrifuged through a genomic DNA removal column. Following this, BSXS buffer was added to the flowthrough and the sample was centrifuged through an RNA retention column. Then, the column was washed with MDB, and then a wash buffer with serial centrifugations. The pellet was reconstituted in RNase-free H_2_O and total RNA was measured using the Quawell Q5000 UV-Vis Spectrometer (Quawell, San Jose, CA, USA).

### 2.13. cDNA Synthesis and qRT-PCR

Two hundred and fifty nanograms (250 ng) of the total RNA from stromal cells, HIOs, and colonoids were reverse-transcribed into cDNA using PrimeScript 1st strand cDNA Synthesis Kit (TaKaRa, Kusatsu, Shiga, Japan), following the manufacturer’s instructions. The mRNA expression of *CCL2* (Table 3) was assessed by quantitative RealTime PCR (qRT-PCR) using Sybr Green (Kapa Biosystems, Wilmington, NC, USA) in SaCycler-96 RUO (Sacace Biotechnologies, Como, Italy), using a two-step amplification protocol. Gene expression was normalized against the expression of the housekeeping gene *GAPDH* in the same sample using the 2^−ΔΔCt^ method.

### 2.14. Enzyme-Linked Immunosorbent Assay (ELISA)

Protein expression of CCL2 in supernatants of stromal cells, HIOs, and colonoids was measured using the Human DuoSet^®^ CCL2 ELISA kit (R&D Systems, Minneapolis, MN, USA), as previously described [34]. Briefly, 96-well plates were coated overnight with capture antibody and then incubated with blocking buffer. Duplicates of each sample, CCL2 standard or blank were pipetted into wells. After 2 h of incubation, wells were washed and then incubated with a biotinylated detection antibody for another 2 h, followed by the incubation with streptavidin conjugated to horseradish peroxidase for 20 min and tetramethylbenzidine with H_2_O_2_ for another 20 min to produce the color of an OD (Optical Density) analogue to the initial CCL2 concentration. ODs were measured at 450 nm on a microplate reader (Diareader EL × 800; Dialab, Wr. Neudorf, Austria). A linear function deduced from the ODs of the standards was used to convert ODs of unknown standards to concentration.

### 2.15. In Vitro Statistics

Statistical analyses for the experiments of cSEMFs, HIOs, and colonoids were performed using Prism Software 10 (GraphPad Software, San Diego, CA, USA, www.graphpad.com accessed on 20 July 2023). The results are presented as the mean with standard error of the mean (SEM) and were analyzed using the Kruskal–Wallis test. Statistical significance was established at an alpha level of *p* < 0.05.

## 3. Results

### 3.1. Myeloid and Stromal Cell Clusters in Inflamed and Non-Inflamed CD Ileum

Preliminary filtering of the 22 ileal single-cell sequenced mucosal biopsies (paired from inflamed and non-inflamed loci of 11 patients) was applied to remove unwanted cell lineages. Cells were reduced from 120,316 to 29,167 and then subjected to UMAP clustering and preliminary characterization using SingleR. Clusters characterized as stromal or myeloid cells were separated and reanalyzed individually and in tandem. The reanalysis resulted in 470 stromal and 1306 myeloid cells for inflamed mucosal biopsies, as well as 702 stromal and 951 myeloid cells for non-inflamed ones. Downstream clustering within these lineages identified six clusters within inflamed and seven within non-inflamed stromal cells, as well as nine within inflamed and eight within non-inflamed myeloid cells.

Although not all genes assessed exhibited similar performance, they provided sufficient information to classify the clusters in accordance with the existing literature [26]. The following markers were most effective for characterizing the stromal 1 fibroblasts (S1): metalloproteases *ADAM28* and *ADAMDEC1* lysing extracellular matrix (ECM), *ZEB2* binding to intracellular SMADs and thus interfering with the profibrotic TGF-β signaling, *BPM4* tagging proteins for proteasomal degradation, chemokines *CXCL14* and *CCL11*, and *CFD* controlling the rate-limiting step of the alternative pathway of complement. A separate set of markers were most effective in characterizing stromal 2 cells (S2): PDGF receptor A *PDGFRA*, cholesterol 5-OHase *CH25H* producing 25HC that has antiviral activity, tissue factor *F3* initiating the extrinsic coagulation pathway, neuropeptide Y *NPY*, a neurotransmitter/modulator, and *BPM4* and *CXCL14* (common with S1). The identification of stromal 3 cells (S3) relied mainly on the marker *KCNN3*, a K^+^/Ca^++^-activated channel, while the stromal 4 (S4) cluster was characterized by *PDPN*, an oncogene, metalloprotease *MMP1*, periostin *POSTN* modifying extracellular matrix scaffold, focal adhesion *FAP* initiating intracellular signaling as an effect of mechanical stimuli, and *TWIST1*, an early development gene. It is also worth mentioning that the S4 population, enriched in inflamed biopsies, presents low expression of several genes also found in non-inflamed S2, S3, and myofibroblasts, suggesting a transitional phenotype. The myofibroblast cluster, on the other hand, was distinguished by the high expression of cytoskeleton- and contractility-related *ACTA2*, *MYH11*, *TAGLN*, and *RSPO*, an embryonic paracrine protein.

Among the myeloid cell clusters, inflammatory macrophages exhibited high expression of the human leukocyte antigen *HLADRB1*, complement proteins *C1QC*, *C1QB*, and the scavenger receptor for haptoglobin–hemoglobin complexes *CD163*. Macrophages shared the first three. LYVE1^+^ macrophages were identified by the presence of *LYVE1* (a membrane receptor for hyaluronan) and *HLADRB1*, while monocytes were characterized by the expression of antibacterial bivalent ion scavengers *S100A4*, *S100A6*, *S100A8*, *S100A9*, antibacterial enzymes lysozyme *LYZ* and myeloperoxidase *MPO*, ficolin 1 *FCN1*, translocator protein *TSPO*, offering protection against intracellular oxidative burst, trophic cytokine *IL1B*, and resistin *RETN*, involved in glucose homeostasis. Finally, conventional dendritic cells (cDC) were distinguished by their high expression of *HLADRB1* and c-lectin *CLEC10A* binding bacterial N-Acetyl galactosamine and initiating phagocytosis, while showing lower expression of *C1QC* and *C1QB*.

Merging clusters with comparable marker expressions was conducted to establish a basis for subsequent analyses. The resultant clusters are depicted in the UMAP plot of Figure 2, providing an insightful depiction of the cellular landscape and highlighting cell-population differences between the sample groups, cellular heterogeneity, spatial organization, and potential cell-to-cell interactions.

### 3.2. Distinct Functions of Stromal and Myeloid Cell Subsets during Homeostasis and Inflammation Outlined by their Transcriptional Profile

We hypothesized that cell populations derived from non-inflamed areas, most probably, would be enriched with transcriptomes involved in homeostatic functions, whereas transcriptomes obtained from inflamed areas would most probably relate to the enhancement or regulation of inflammatory pathways. Therefore, differentially expressed genes among clusters of stromal and myeloid cells in the non-inflamed area may help identify the distinct roles undertaken by different cell subsets during homeostasis, while the same investigation in the inflamed area could elucidate subset-distinct roles in inflammatory processes.

S1 fibroblasts in non-inflamed areas preferentially expressed genes associated with the maintenance of the intestinal barrier (*ADAMDEC1*), innate immunity against pathogens (*CCL11*, *CCL8*, *CCL13*), antigen presentation of lipoproteins [35] (*APOE)*, and ECM remodeling and cell adhesion (*A2M*, *LUM*, *FBLN1*, *DCN*) (Figure 3). In sharp contrast, the transcriptional program of the S1 in the inflamed mucosa is dominated, compared to other subsets from the same area, by chemotactic signals to monocytes, neutrophils and B-cells (*CXCL14*, *MT-ND3*), angiogenesis (*EDIL3*), and suppression of complement activation (*CFH*) (Figure 4). S2 stromal cells—also referred to as telocytes—in the non-inflamed tissue preferentially expressed genes to support epithelial proliferation, differentiation, and survival (*RGS5*, *FRZB*, *CAV1*, *GADD45B*) and genes associated with the cytoskeleton (*CALD1*) (Figure 3). S2 cells from inflamed tissue, compared to other inflamed stromal subsets, preferentially expressed genes involved in antigen presentation (*HLA-DRA*, *CD74*), humoral immunity (*IGHA1*), extracellular tissue remodeling that facilitates neutrophil migration (*SPP1*), Ca^++^ scavengers (*S100B*), and chemokine receptors (*CXCR4*) (Figure 4).

Due to our non-integration strategy, we were able to identify the S3 stromal cells solely in non-inflamed tissue by the unique expression of KCNN3, which suggests that these cells are of pluripotent nature and give rise to other stromal cell subtypes in an inflammatory environment such as S4. Both S3 and non-inflamed S4 exhibited a similar transcriptomic signature involving genes such as heat shock protein *CRYAB*, chaperone *CLU*, *tetraspanin CD9*, *ALDHA1A*, and lipoprotein *PLP1* (Figure 3). Their transcriptional program suggests their participation in immunoregulation through the regulation of chaperone activity (*CRYAB*, *CLU*) and ECM organization (*SPARC*). Specifically, for S4, transcriptome in non-inflamed areas was enriched in genes involved in antigen presentation (*CD74*, *HLA-DRA*, *HLA-DRB1*), whereas in inflamed areas S4 stromal cells assumed a more polarized state guided towards ECM deposition and remodeling: in detail, inflamed S4 expressed collagen genes (*COL3A1*, *COL1A1*, *COL1A2*), matrix remodeling genes (collagen triple helix repeat containing 1 *CTHRC1*, cathepsin K *CTSK*), and lectin *CHI3L1* that binds chitin on bacterial walls (Figure 4). Myofibroblasts from non-inflamed areas expressed cytoskeleton and contractility genes (*ACTA2*, *ACTG2*, *MYH11*, *MYLK*, *TAGLN*), whereas myofibroblasts from inflamed areas increased expression of myosin light chain 9 (*MYL9*) and genes involved in basement membrane repair (*COL4A1*) (Figure 3 and Figure 4).

The DC transcriptomic program in areas that were not inflamed was dominated by genes associated with baseline lipid metabolism (*FABP5*), antigen phagocytosis (*CAV1*), sensing and tolerance (*PLVAP1* plasmacytoid DC marker), and support of B-cell proliferation/differentiation to plasma cells (*CD320*) (Figure 5). In contrast, the transcriptional program of DCs in inflamed areas involved antigen presentation (*HLA-DPB1*) and intestinal barrier support (amphiregulin *AREG* binding on epithelial growth factor receptor) (Figure 6). LYVE1^+^ macrophages in inflamed and non-inflamed areas shared common expression of genes that regulate trafficking of leukocytes via cytokines (*CCL21*), adhesion proteins (*MMRN1*, *LYVE1*), inhibitors of ECM lysis (inhibitor of metalloproteinases *TIMP3*), and the homeostatic long non-coding RNA (lncRNA) *MALAT1*. Interestingly, they also expressed the *NEAT1* lncRNA that activates inflammasomes. On top of the aforementioned commonly expressed genes, inflamed LYVE1^+^ macrophages express higher levels of profibrotic genes like *IGF2* and *EFEMP1* (Figure 4 and Figure 5). Monocytes from non-inflamed areas preferentially expressed genes of cytokines (*IL1B*), chemokines (*CXCL2*), and NF-κΒ inhibitors (*NFKBIA*), whereas monocytes from inflamed areas, exhibited a wider spectrum of chemokines (*CCL3*, *CCL4*, *CCL3L3*) and Fe scavengers (*MT1G*, *MT2A*) (Figure 5 and Figure 6).

Non-inflamed macrophages express an array of canonical (*HLA-DPB1/DRA/DRB1*) and non-canonical (*HLA-DQA1*) major histocompatibility complex (MHC) class II molecules and associated genes (*CD74*), along with complement genes (*C1QA*, *C1QB*). They also express cystatin C (CST3), a cysteine protease (Figure 5). Inflamed macrophages adopt a more active and complex role that involves Fe sequestration (ferroportin *SLC40A1*), cytokinesis (septin *SEEP1*), expression of complement components (*C1QA*, *C1QB*, *C1QC*), innate immune defense by enzymes (*LYZ*), and suppression of the opposing phenotype of tolerogenic (M2) macrophages (*SEPP1*) (Figure 6). Inflammatory macrophages were only detected in inflamed loci and preferentially expressed an array of proteases (cathepsins *CTSD*, *CTSB*), an inducible metalloprotease to lyse ECM (*MMP9*) and a T-cell attracting chemokine (*CCL18*) (Figure 6).

Figure 3, Figure 4, Figure 5 and Figure 6 summarize the transcriptional variation among different clusters within the same cell lineage in the inflamed and non-inflamed terminal ileum of CD patients and depict the key genes associated with each cluster.

### 3.3. Communication Pathways between Myeloid and Stromal Cells

The CellChat analysis revealed several, inferred by transcription, communication pathways that are likely to occur between different clusters of stromal and myeloid cells. In non-inflamed areas, it was observed that mostly S1 and S3 act as the source cells while monocytes and macrophages function as responders. In contrast, in inflamed areas, communication between S4 and inflammatory macrophages prevailed. The chord diagrams of Figure 7 provide a comprehensive overview of the intercellular communication patterns, highlighting both the quantity and quality of information exchange among different cell types. Further analysis detected possible interactions of collagen types *COL1A1*, *COL1A2*, *COL4A1*, *COL4A2*, *COL6A1*, *COL6A2*, and *COL6A3* with the *CD44* receptor suggesting an ECM-mediated communication between S4 and inflammatory macrophages in inflamed areas and between S3 and monocytes in non-inflamed areas (Figure 8). Other ECM-related genes participating in possible crosstalk between stromal cells and monocytes/macrophages included fibronectin (*FN1*) and laminins (*LAMA4*, *LAMB1*), interacting with the same receptor on monocytes and macrophages. Additionally, the amyloid precursor protein (*APP*) interacting with the *CD74* receptor was another signaling molecule between inflamed LYVE1^+^ macrophages and various immune cells, as well as between non-inflamed S3 stromal cells and macrophages or monocytes (Figure 8).

*CD99*, a multifunctional membrane protein, was found in both outgoing and incoming signals to S4 in inflamed, as well as in the outgoing and incoming signals of myofibroblasts, S2, and S3 in non-inflamed samples. Chemokine *CXCL12* was also identified as a potential outgoing signal from inflamed S1 and S4 to DCs and macrophages through the *CXCR4* receptor of the latter. In non-inflamed loci, S1 were identified as the source and monocytes and macrophages as the recipients (Figure 8).

Two pathways appeared exclusively in non-inflamed areas: (a) osteopontin (*SPP1*) by S3 and S4 to *CD44* on monocytes and (b) annexin A1 (*ANXA1*) by S3 to formyl peptide receptor 1 (*FPR1*) on monocytes. More pathways were identified solely in inflamed loci. In one such pathway, S4 stromal cells transcribed the macrophage migration inhibitory factor (*MIF*), and macrophages, DCs, inflammatory macrophages, and monocytes were the recipients via the *CD44*, *CD74*, and *CXCR4* receptors. Another interaction was observed between S4 stromal cells transcribing midkine (*MK*) and possible recipients included S1 fibroblasts and LYVE1^+^ macrophages via the NCL receptor as well as inflammatory macrophages and myofibroblasts via the *SDC2* receptor. Pleiotrophin (*PTN*) was another outgoing signal from S1 to the same recipients. Lastly, thrombospondin (*THBS*)-1 and -2 functioned as outgoing signals, primarily from S4 stromal cells, with LYVE1^+^ macrophages, macrophages, and myofibroblasts being the main recipients through the *CD36* receptor (Figure 8).

### 3.4. Interactions between Stromal and Myeloid Cells Regulate Pathogenic Chemokines and Cytokines

To gain a more in-depth understanding of the intercellular communications network and examine its effects on crucial chemokines and cytokines, we adopted the NicheNet approach. This enabled the investigation of transcriptomic alterations induced by ligand–receptor interactions on target cell populations. In the inflamed terminal ileum, *TNF-α* signaling on S1, S2, S4, and myofibroblasts emerged with a multitude of targets (Figure 9A–D). Monocytes were identified as the primary *TNF-α* source. Various chemokines (*CCL11*, *CCL17*, *CCL19*, *CCL20*, *CCL27*, *CCL2*, *CCL3*, *CCL4*, *CCL5*, *CCL7*, *CCL8*, *CCL22*, *CXCL2*, *CXCL3*, *CXCL5*, *CXCL9*, *CXCL10*, *CXCL11*, *CXCL12*, *CXCL16*, *CXCL23*, *CXC3L1*, and *XCL1*), interleukins (proinflammatory *IL-11*, *IL-12*, *IL-13*, *IL-18*, *IL-24*, *IL-1A*, and its decoy receptor; immunoregulatory *IL-10)*, and two growth factors (*CSF-1*, *-2*) were affected by this pathway. *IL-1β*, predominant in the non-inflamed mucosa, retained some significance in the inflamed mucosa too for signaling from monocytes to S1 and S4 stromal cells (Figure 9A,C and Figure 10A,D). Importantly, during inflammation *IL-1β* was the only molecule projected in interacting with key chemokines (*CXCL1*), cytokines (*IFN-γ*, *IL-15*, *IL-17*), and *TGF-β1*. The monocyte-derived *OSM* (oncostatin M of the *IL-6* family) impacted stromal cells and myofibroblasts in inflamed regions, resulting in the transcription of *IL-6* and *CCL2* (Figure 9A–D). *IL-6*, produced by monocytes in inflamed areas, exhibited common targets (*CCL2*, *IL-4*, *IL-11*, *IL-12*, *IL-17*, *TGF-β1*, *IL-10*) in S1, S2, and myofibroblasts (Figure 9A,B,D). *IL-1α* expression by monocytes in inflamed areas affected the transcription of *CCL2*, *CCL8*, *CXCL1*, *CXCL3*, *CXCL10*, and *IL-1β* on S1 fibroblasts (Figure 9A). *IL-33*, generated by LYVE1^+^ macrophages in inflamed areas, exclusively targeted *CXCL1* transcription in S1 fibroblasts (Figure 9A). *TGF-β1* signaling from macrophages primarily targeted key cytokines (*IFN-γ*, *IL-17A*, *IL-4*, *IL-12*) and a few chemokines in all inflamed stromal cell subtypes (S1, S2, S4) and myofibroblasts (Figure 9A–D).

Interestingly, the analysis revealed a few stimuli, so far underappreciated for CD, that are involved in various signaling networks. One of them is high mobility group box 1 protein (*HMGB1*), a neuro-inflammatory cytokine, originating from inflamed LYVE1^+^ macrophages, myofibroblasts, and S1, S2 stromal cells, specifically affecting the transcription of *CCL2*, *CCL5*, and *IL-18* in S4 cells and myofibroblasts (Figure 9C,D). *APOE* was verified by the NicheNet approach too, as a significant signal for all stromal cells and myofibroblasts in inflamed loci. Inflammatory macrophages were its main source in the inflamed tissue (Figure 6A–D), while this role was undertaken by S1 and myofibroblasts in the non-inflamed terminal ileum (Figure 10A,E). The source of galectin 3 (*LGALS3*) in the inflamed terminal ileum shifted from myofibroblasts to inflammatory macrophages and *CCL20* became the specific target in S1, S2, S4 and myofibroblasts (Figure 9). MIF from S4 targeted the transcription of chemokines (*CXCL5*, *CXCL1*, *CXC3L1*), as well as interleukins (*IL-11*, *IL-12*, *IL-7*) in myofibroblasts (Figure 9D). Unexpectedly, an inducible metalloprotease (*MMP-9*) from the inflammatory macrophages appears to modify *CXCL1* transcription while *Jagged-1*, a membrane protein initiating the NOTCH pathway, targeted a few chemokines and *IFN-α* and *IL-17b* in S2 (Figure 9B).

As was the case for stromal cells, *TNF-α* and *IL-1β* from infiltrating monocytes were the primary inputs for DCs, inflammatory macrophages, LYVE1^+^ macrophages, and monocytes in inflamed areas, leading to transcriptional modifications of a wide range of cytokines and chemokines (Figure 11A–E). Moreover, ADAM metallopeptidase domain 17 (*ADAM17*) from LYVE1^+^ macrophages and *ICAM-1* from monocytes exhibited specific effects on inflamed DCs, such as transcriptional modifications of *IL-17*, *IL-18*, *TNF-α*, and *IFN-γ*, *IL-4*, *IL-5*, respectively (Figure 11A). *HMGB1* from LYVE1^+^ macrophages, myofibroblasts, and S1 fibroblasts had a more limited repertoire of effects on inflamed DCs (*CCL2*, *CCL5*, *IL-6*, *TNF-α*), with no differences between inflamed and uninflamed loci (Figure 11A and Figure 12A). In all myeloid cells, *TGF-β1* from macrophages affected a few chemokines and, importantly, the cytokines *IFN-γ* and *IL-17*. Additionally, locus non-specific effects were observed for *LGALS3* by inflammatory macrophages, exclusively targeting *CCL20*, and for MMP-9, exclusively targeting *CXCL1* (Figure 11).

The most prominent signals from stromal cells to myeloid cells in the inflamed areas included *CCL2* and *MIF. CCL2* emerged as a signal deriving from S1 to macrophages of the inflamed mucosa, with transcriptional effects on *CCL3-5*, *TGF-β1* and *IL-13* (Figure 11C)*. MIF* expression was prominent in inflamed S4 cells affecting dendritic cells, macrophages, monocytes, and inflammatory macrophages while its transcriptional targets included *CXCL1*, CXCL5, *CX3CL1*, *IL-7*, *IL-9*, *IL-11,* and *IL-12A* (Figure 11A,C–E).

To summarize, NicheNet delineated a wealth of specific or redundant signaling networks between clusters of stromal and myeloid cells in the terminal ileum of CD patients. A sizable number of these networks were specific for inflamed loci, and novel, non-reported so far interactive pathways for ileal CD emerged.

### 3.5. CCL2 Is Upregulated in Ileal CD and Produced by Stromal Cells in Response to Proinflammatory Stimuli

Among others, our analysis featured *CCL2*, a chemoattractant chemokine in S1 fibroblasts with *IL1*, *IL-6*, *OSM*, and *TGF-β* affecting its transcription. In our recent study [34], *CCL2* was identified as a central (hub) gene in inflammatory and profibrotic processes. Reanalyzing the raw data from this study, *CCL2* was overexpressed six-fold and was positioned within the top 20 overexpressed genes in ileal CD versus healthy individuals (Figure 13A,B).

We then studied *CCL2* regulation by intestinal stroma via a series of wet lab experiments. To explore the stromal contribution to CCL2 production, we compared CCL2 expression from intestinal organoids (HIOs) and colonoids. HIOs consist of both epithelial and stromal cells, whereas colonoids consist of only epithelial cells; therefore, differences in CCL2 production might be attributed to the contribution of the stromal component. Unstimulated organoids and colonoids had minimal mRNA and protein expression of CCL2 (Figure 13C,D) suggesting that CCL2 is only produced in the context of an inflammatory microenvironment. Indeed, treatment with recombinant IL-1α and TNF-α upregulated mRNA and protein of CCL2 in both HIOs (mRNA: 52.3-fold, ±2.6, *p* < 0.0001; protein: 3.5 ± 0.2 ng/mL, *p* < 0.0001; Figure 13C,D) and colonoids (mRNA: 39.4-fold, ±5.86, *p* < 0.001; protein: 2.6 ± 0.4 ng/mL, *p* < 0.001; Figure 13C,D). The fact that the effect of these cytokines was more prominent in HIOs (*p* < 0.05; Figure 13C,D) indicates that stromal cells might be an important source of *CCL2* in the inflamed state of the intestinal mucosa.

Then we assessed *CCL2* transcription and secretion directly by primary human intestinal stromal cells isolated from the ileum of healthy individuals or patients with CD. Quiescent stromal cells from CD patients or healthy subjects in culture exhibited no differences in *CCL2* expression or production (Figure 8E,F). Treatment with proinflammatory cytokines induced a robust CCL2 response by isolated intestinal stromal cells thus confirming that the inflammatory microenvironment is capable of promoting strong CCL2 responses by intestinal stroma. However, we failed to identify differences in *CCL2* transcription and secretion between CD and healthy stromal cells following treatment with proinflammatory cytokines. Specifically, IL-1α and TNF-α treatment equally induced CCL2 mRNA and protein in healthy (mRNA: 75.2-fold, ±28.9, *p* < 0.05; protein: 40.9 ± 10.7 ng/mL, *p* < 0.05) and CD (mRNA: 106.4-fold, ±46.4, *p* < 0.05; protein: 37.8 ± 12.2 ng/mL, *p* < 0.01) stromal cells (Figure 13E,F). In contrast, stromal cells isolated from healthy individuals stimulated with tissue culture supernatants from biopsies of either healthy controls or patients with CD upregulated *CCL2* mRNA (7.7-fold, ±2.2, *p* < 0.05 and 15.6-fold, ±2.2, *p* < 0.001, respectively; Figure 13G), with CD supernatants tending to be more efficient inducers of *CCL2* expression. The above findings indicate that an inflammatory milieu is required to reproduce stromal chemokine responses in vitro and in this context the CD-associated inflammatory microenvironment is a more potent inducer of *CCL2* by intestinal stromal cells.

## 4. Discussion

In this study, we elucidated the interactions between stromal and myeloid cell populations that lead to inflammation and ECM derangement in CD. In terminal ileum biopsies of CD patients, we identified five distinct stromal cell populations, namely S1–S4 stromal cells and myofibroblasts, and five myeloid cell populations, namely DCs, monocytes, macrophages, LYVE1^+^ macrophages, and inflammatory macrophages. These clusters, defined by their gene expression signatures, have discrete communication pathways, in between and among clusters, and differences in transcriptional programs related to the inflammation status of the area they originate from. More importantly, we compared the transcriptional programs of each cluster in inflamed and non-inflamed CD loci to delineate differences in cell–cell communication signaling pathways in homeostasis and inflammation in CD. Lastly, we verified with wet lab experiments on organoids and primary intestinal stromal cells that the stroma is an important source of CCL2 in ileal CD as indicated by the in silico findings.

In non-inflamed areas, S1 contributed to the intestinal immune barrier via protective molecules, such as metalloproteinase *ADAMDEC1* [36] and several C-C motif chemokines involved in protective innate immune responses [37]. Implied by the differential expression of *RGS5* and *FRZB*, S2 most likely represent human crypt-to-top fibroblasts lining the inner surface of the basement membrane and guiding epithelial cell differentiation and renewal [38]. S3 had a very limited differential expression repertoire, with the Aldehyde Dehydrogenase 1 Family Member A1 gene (ALDH1A1) spiking out, and the cluster turned undetectable in inflamed loci. This suggests a pluripotent potential of the S3 cluster giving rise to other phenotypes of mesenchymal cells [39], as also reported by Kichen et al. [40] and indicated by the similarities in the transcriptional program of S3 and S4.

DCs of non-inflamed areas support and organize the growth of B-cells in germinal centers through *CD320* [41], adopt a tolerogenic phenotype via expression of FABP5, a homeostatic chaperone that is known to suppress cytokines and chemokines [42], and are involved in antigen sampling, as denoted by the mRNA expression of caveolin-1 and *PLVAP1* [43]. The differential expression profile of LYVE1^+^ macrophages assigns them roles in organizing leukocyte migration to the intestinal mucosa with chemokines and ECM molecules. CCL21 recruits eosinophils [37] and DCs [44]. Multimerin-1 (MMRN1), an ECM protein [45], binds activated neutrophils via interaction with the ανβ3 integrin [46]. Interestingly, non-inflamed LYVE1^+^ macrophages preferentially expressed the long non-coding RNA gene *NEAT1*, for which the only relevant role recognized so far has been macrophage polarization to M1 [47]. Macrophages function as antigen-presenting cells orchestrating immune responses and carry an armamentarium of direct cytotoxicity weapons (*HLA-DPB1*/*DRA*/*DRB1*/*DQA1*, *CD74*, and *C1QA*/*B*, respectively). They also expressed cystatin C (*CST3*), a cysteine protease inhibitor, shown to inhibit *IL-1b* and *TNF-α* effects on activated monocytes of CD patients [48]. Other immunoregulatory molecules included *IκΒα* inhibitor (*NFKBIA*) and *CXCL2*, which have been shown to suppress inflammatory responses and development of experimental colitis [49].

Inflamed tissue was enriched in stromal cells with enhanced transcription of ECM building blocks and inflammatory macrophages with increased gene expression of protease, ECM and T-cell attracting chemokine genes. Activated S1 fibroblasts differentially expressed *CXCL14*, a chemoattractant chemokine for neutrophils, monocytes, DCs, B-cells, and NK cells [50]. S4 actively contribute to ECM deposition and remodeling, being enriched in collagen type III (*COL3A1*), type I (*COL1A1*, *COL1A2*), and *CTHRC1* expression [51]. Inflammatory macrophages of the inflamed tissue differentially expressed proteolytic enzymes cathepsin D and B (*CTSD*, *CTSB*) and matrix metalloproteinase 9 (*MMP9*), as well as the T-cell *CCL18* chemokine [52]. Inhibition of both *CTSB* and *CTSD* has been shown to ameliorate DSS colitis [53] and overexpression of *MMP-9* to aggravate DSS and infectious colitis [54].

We showed intense communication between stromal and myeloid cells in both inflamed and non-inflamed samples, mainly through ECM molecules and chemokines resulting in retention of monocytes and inflammatory macrophages, and immune cell polarization and differentiation. ECM molecules, including collagens, fibronectin, and laminins, hold significant roles in macrophage and monocyte infiltration [55]. *FN1* is of particular interest and aggregates of it promote a distinct but mixed pro- and anti-inflammatory phenotype on macrophages [56]. In contrast, in homeostasis, *FN1* attracts fibrinolytic monocytes by means of matrix metalloproteinase 7 [57]. Regarding chemokines, *CXCL12* holds a dual role: a potent acute phase inflammation chemoattractant for monocytes, also conferring chronic inflammation and fibrosis via lymphocyte and macrophage chemotaxis [58]. *APP* and *CD99* are also involved in these stromal–myeloid communications in both inflamed and non-inflamed samples. Accordingly, *APP*^−/−^ mice had reduced macrophage infiltration and activity in their ilea [59], suggesting that *APP* is indeed implicated in the recruitment of macrophages. Interestingly, *CD99* regulates monocyte-to-DC differentiation towards pro-inflammatory *IL-12* and *CCL-1* producers by downregulating *CD1A* expression [60]. To this notion, the increased expression of *MIF* we found in inflamed samples only has been correlated with numerous autoimmune diseases, such as IBD [61], and by binding to different receptor complexes, such as *CD74/CD44*, *CD74/CXCR2*, *CD74/CXCR4*, and *CD74/CXCR4/CXCR7*, the resulting outcome may differ [62], suggesting that it might contribute to time- and space-dependent differential activation of immune cells.

We assembled and prioritized, between the wealth of signaling pathways identified, cells producing signaling molecules, responder cells, and transcriptional programs activated. TNF-α signaling from monocytes to all S1, S2, and S4 cells of inflamed areas induced a battery of chemokines, the pan-T cell trophic cytokine IL-1β, as well as Th1-polarizing *IL-12* and Th2-polarizing *IL-13* cytokines. However, in inflamed mucosa, IL-1β from monocytes was unique in inducing *IL-15* in S4 and this interleukin is unique in maintaining the *niche* of the natural killer cells [63]. Of note, all stromal clusters secreted *IL-7*, which regulates the development and homeostasis of immune cells, including T lymphocytes, B lymphocytes, natural killer cells, and innate lymphoid cells [64]. *HMGB1*, a neuro-inflammatory cytokine, minimally investigated so far, originated from LYVE1^+^ macrophages, myofibroblasts, and S1, S2 cells, and was quite specific in inducing *CCL2* and *CCL5* in S4, myofibroblasts and DCs, *IL-18* in S4 and myofibroblasts, and *IL-6, TNF-α* on DCs. Of note, it has also been proposed as a fecal biomarker of intestinal inflammation in CD [65].

Most importantly, we showed that communication between stromal and myeloid cells strengthens pro-inflammatory amplification loops, which may contribute significantly to the development of ileal CD. In detail, S1 in inflamed areas upregulate *CCL2* by integrating *TNF-α*, *OSM*, *IL-6*, *IL-1α*, and *IL-1β* signaling from a multitude of sources, mainly myeloid cells. All of these cytokines have been linked to the enhancement of stromal chemokine gradients, which is proposed as a mechanism contributing to the development of Crohn’s disease and resistance to treatment [21,66]. Increased *CCL2* transcripts have been observed in the lamina propria of IBD patients, and *CCL2* serum levels have been associated with treatment failure of anti-TNF-α and anti-p40 agents in CD [67]. *CCR2* (i.e., *CCL2* receptor) on *Ly6C*^hi^ monocytes recruits them to the inflamed intestinal lamina propria in experimental colitis models, where they become the predominant mononuclear cell cluster. To this point, the elimination of circulating monocytes with anti-*CCR2* antibodies prevents severe disease [68]. In human IBD, the expansion of pathogenic pro-inflammatory *CD11c^hi^CCR2^+^CX3CR1^+^* macrophages is driven by enhanced *CCL2* gradients and contributes to colonic inflammation during IBD [69]. Kamada et al. identified a myeloid subset expanded in human CD expressing both macrophage and DC markers, as well as *CX3R1* and *CCR2*. These cells produce higher levels of proinflammatory cytokines, such as *IL-23*, *TNFα*, and *IL-6*. This abnormal macrophage differentiation is triggered by *IFN-γ* produced by lamina propria macrophages and is associated with granuloma formation in mice [70]. Interestingly, Myscore et al. reported that *CCL2* not only recruits monocytes but also induces transcriptomic changes that drive their differentiation into pro-inflammatory immature macrophages in renal inflammation [71].

Based on our findings, we propose that these pathogenic inflammatory macrophages or monocyte-like cells are equivalent to the inflammatory macrophages we identified through the single-cell transcriptomic analysis of inflamed ileal CD samples. They respond to *CCL2* signals from S1 fibroblasts for their recruitment and receive signals through *IFN-γ* and *TNFα* from S4 stromal cells and other myeloid cells in the lamina propria for their differentiation (graphical abstract). *IFN-γ* has been identified as a major inducer of pro-inflammatory macrophage differentiation [72]. Signaling from ECM proteins may also contribute to the development of the pro-inflammatory macrophage phenotype. Engagement of CD44 on human macrophages has been shown to enhance nuclear translocation of NF-κB and downstream proinflammatory cytokines, such as *IL-1β* and *TNFα* [73].

*FN1* has been identified as a potential ligand capable of eliciting similar responses [74]. Importantly, fibronectin-mediated cell adhesion was necessary for the induction of *MMP-9* simultaneously to macrophage differentiation [75]. Thus, the outgoing signals from S4 cells, including *FN1* and collagens, interacting with *CD44* on inflammatory macrophages, have the potential to regulate the pro-inflammatory differentiation programming in macrophages. Lastly, *MIF* is another prominent communication pathway between stromal and myeloid cells in inflamed areas identified by our study. *MIF* is a potent inducer of inflammatory responses participating in innate immunity and various autoimmune diseases [76]. We demonstrated that it is transcribed in S4, as well as monocytes, macrophages, and inflammatory macrophages. This suggests that stromal-derived MIF may also participate in ileal inflammation in CD [77].

Having identified stromal *CCL2* as a prominent communication pathway between stromal cells and myeloid cells in ileal CD we confirmed *CCL2* overexpression in ileal samples of patients with CD and explored its stromal origin and regulation by TNFα and IL-1α with a series of in vitro studies. Our prior studies have demonstrated that the presence of mesenchymal cells in organoids was required for the production of *CCL2* and that pro-inflammatory cytokines upregulated it [34]. In the present study, we also show that colonoids, which only contain epithelial cells [78], can produce CCL2 upon stimulation but maximal upregulation can only be achieved in organoids that harbor stromal cells. The fact that isolated stromal cells from CD patients or normal controls can produce CCL2 in vitro upon stimulation with proinflammatory cytokines but maximal expression was achieved via stimulation with supernatants of CD intestinal tissue, strongly implicates the inflamed CD microenvironment in mediating stromal CCL2 upregulation.

## 5. Conclusions

In conclusion, single-cell RNA sequencing data, from inflamed and non-inflamed ileal mucosa of CD, enabled us to gain insights into the functional specialization of stromal and myeloid cell clusters and revealed changes in phenotype and function from the assumed homeostatic state of the non-inflamed mucosa to that of the inflamed one. Computational analysis outlined the most probable interacting pathways that operate in ileal CD. We outlined a pathway dependent on CCL2, confirmed its relevance to ileal CD, and reported on its regulation using intestinal stromal cells. Our study indicates that stromal cell-derived CCL2 may sustain small-intestinal inflammation in CD and could represent a valuable therapeutic target.

## Figures and Tables

**Figure 1 biomedicines-12-01674-f001:**
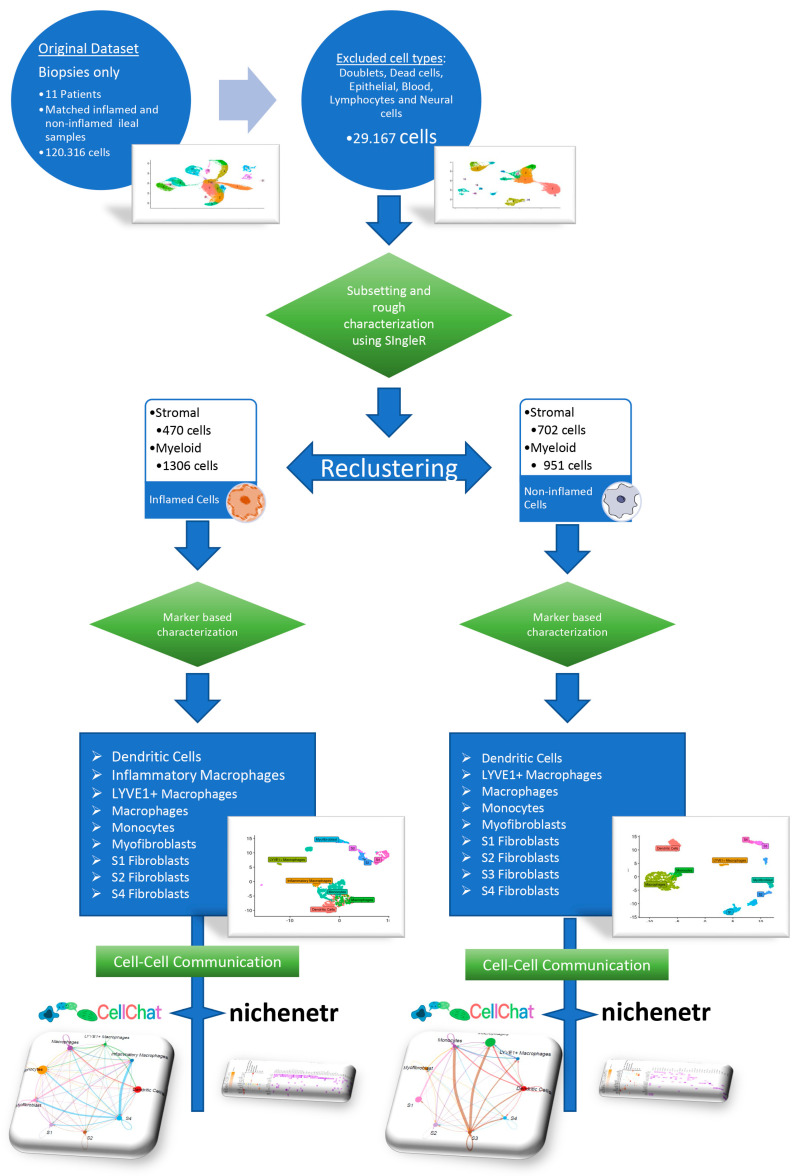
Illustration of the pipeline followed for analyzing the single-cell RNA-sequencing data. The pipeline, presented extensively in the materials and methods section, encompasses several stages, including cell population filtering, clustering, annotation, and investigation of cell–cell communication.

**Figure 2 biomedicines-12-01674-f002:**
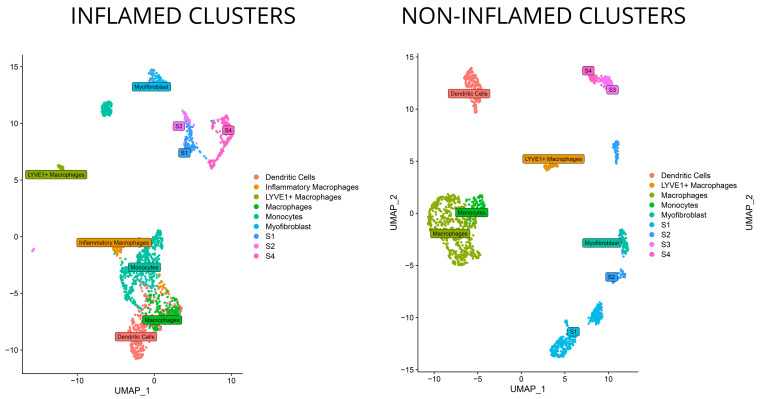
DimPlot visualizations showcasing the UMAP (Uniform Manifold Approximation and Projection) embedding of the single-cell RNA sequencing (scRNA-seq) data from endoscopic biopsies from inflamed or non-inflamed mucosa of the terminal ileum of patients with Crohn’s disease. Each dot represents a single cell, and the positions of the dots reflect their low-dimensional representations generated by UMAP. Clustering analysis was conducted, and a distinct color was assigned to each cluster.

**Figure 3 biomedicines-12-01674-f003:**
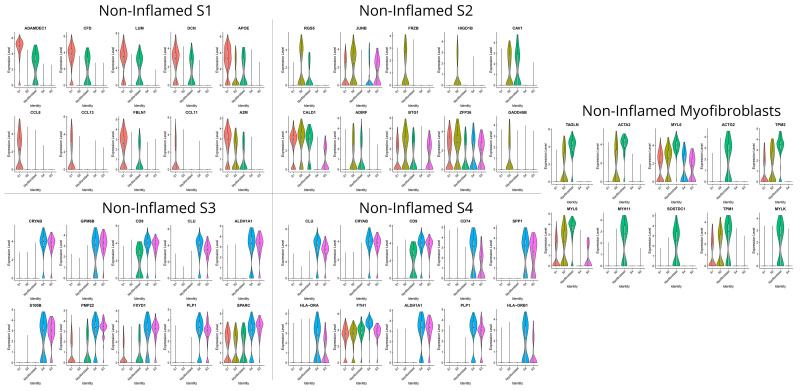
Violin plots that display the expression levels of differentially expressed genes (DEGs) in each non-inflamed stromal cluster, as identified with the Seurat analysis framework. Each violin plot represents the distribution of relative gene expression in the cluster versus all other clusters. The width of the violin depicts the density of cells at each expression level, while the white dot within the violin denotes the median expression value.

**Figure 4 biomedicines-12-01674-f004:**
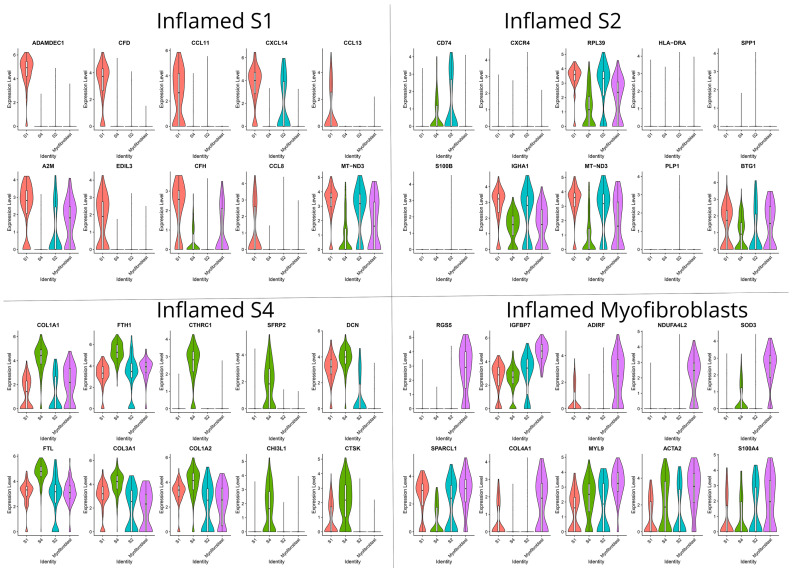
Violin plots that display the expression levels of differentially expressed genes (DEGs) in each inflamed stromal cluster, as identified with the Seurat analysis framework. Each violin plot represents the distribution of relative gene expression in the cluster versus all other clusters. The width of the violin depicts the density of cells at each expression level, while the white dot within the violin denotes the median expression value.

**Figure 5 biomedicines-12-01674-f005:**
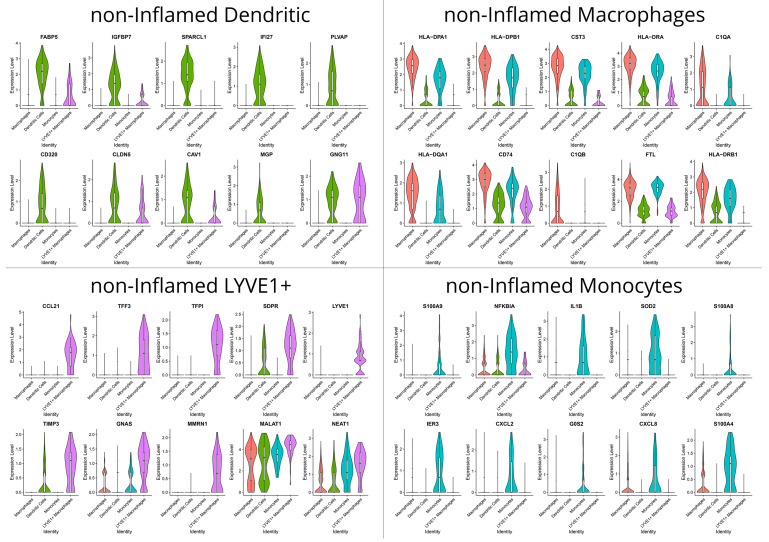
Violin plots that display the expression levels of differentially expressed genes (DEGs) in non-inflamed myeloid clusters, as identified with the Seurat analysis framework. Each violin plot represents the distribution of relative gene expression in the cluster versus all other clusters. The width of the violin depicts the density of cells at each expression level, while the white dot within the violin denotes the median expression value.

**Figure 6 biomedicines-12-01674-f006:**
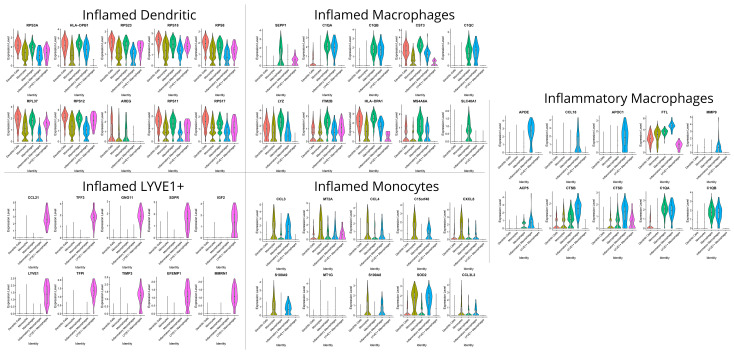
Violin plots that display the expression levels of differentially expressed genes (DEGs) in inflamed myeloid clusters, as identified with the Seurat analysis framework. Each violin plot represents the distribution of relative gene expression in the cluster versus all other clusters. The width of the violin depicts the density of cells at each expression level, while the white dot within the violin denotes the median expression value.

**Figure 7 biomedicines-12-01674-f007:**
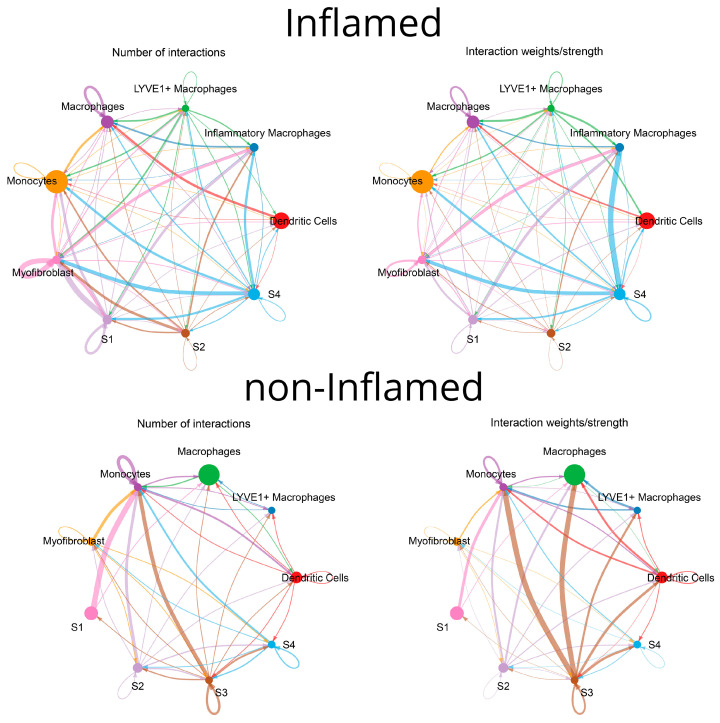
Chord diagrams generated by CellChat, depicting the communication landscape between and across the different clusters of myeloid and stromal cells in inflamed and non-inflamed Crohn’s disease terminal ileum. In the “number of interactions” diagrams, the width of the chords corresponds to the number of interactions between cell clusters. In the “interaction weights/strength” diagram, the thickness of the chords conveys the significance of the interactions based on the probability of an interaction occurring. All interactions are directed from the cell type which expresses the ligand to the one expressing the receptor.

**Figure 8 biomedicines-12-01674-f008:**
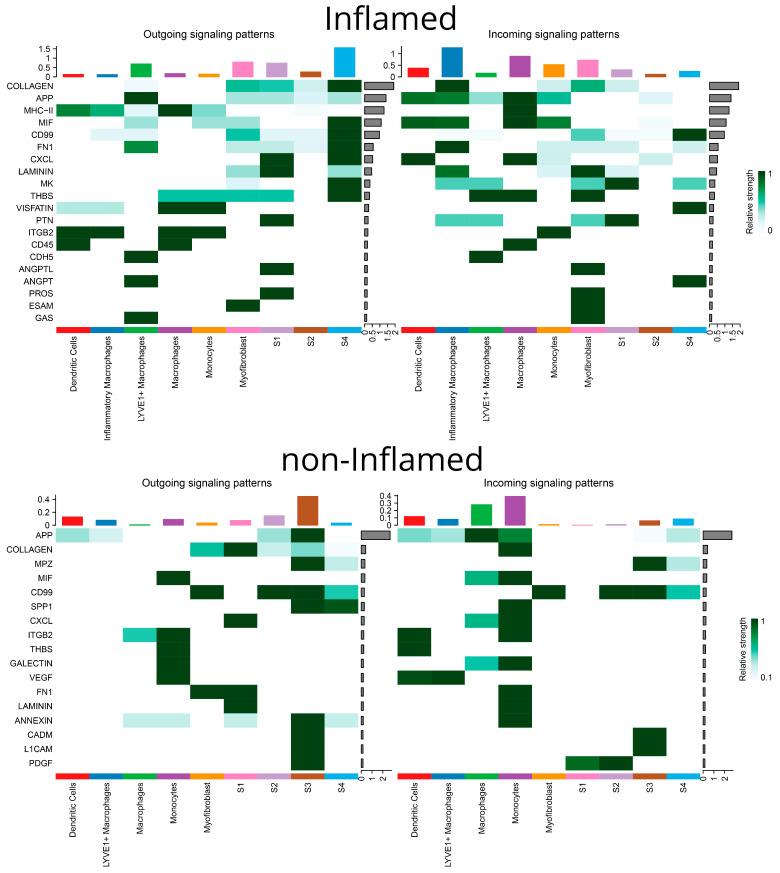
A visualization generated by CellChat, illustrating the outgoing and incoming signaling patterns facilitated by specific pathways represented on the Y-axis. The opacity of the green squares signifies the relative strength (probability) of interaction. The top bars, colored with a representative color for each cell type, depict the number of pathways each cell type sends or receives.

**Figure 9 biomedicines-12-01674-f009:**
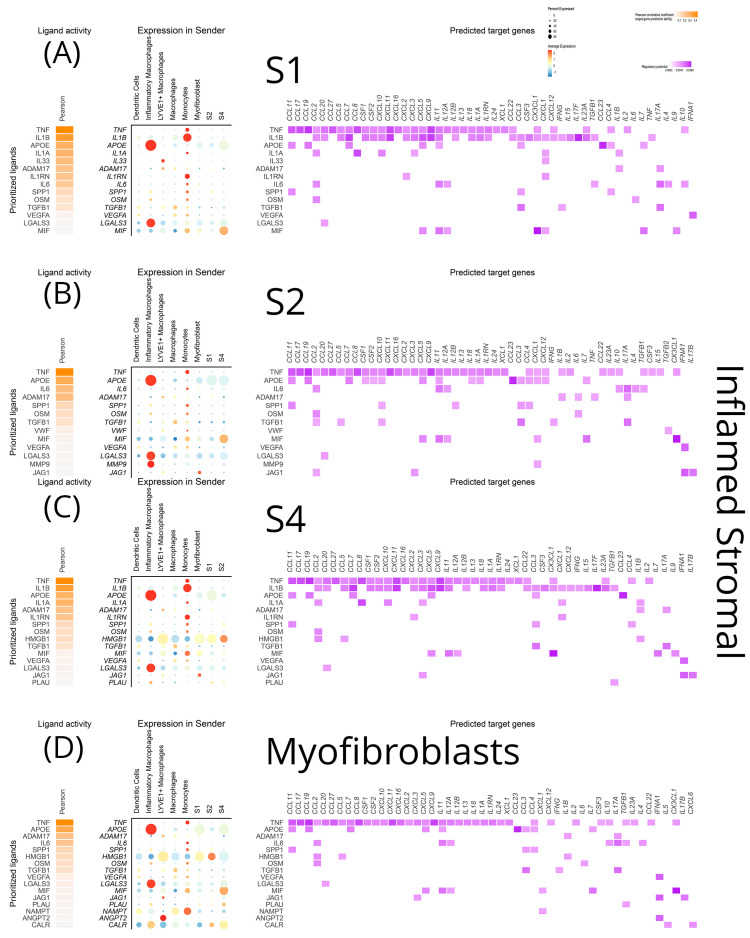
NicheNet visualization of a predictive analysis that establishes connections between cell sources and targets of ligands, as well as effects of the latter on the transcriptional program of the target cells. All stromal clusters were analyzed as both targets and sources of several cytokines, chemokines, growth factors, and signaling molecules within their inflammation status. In each diagram for each stromal cluster, from left to right we depict the top ligands affecting it, their sources, and their effects on the transcriptional program of the target cluster. These plots showcase the signals received by (**A**) inflamed S1, (**B**) inflamed S2, (**C**) inflamed S4, (**D**) inflamed myofibroblasts

**Figure 10 biomedicines-12-01674-f010:**
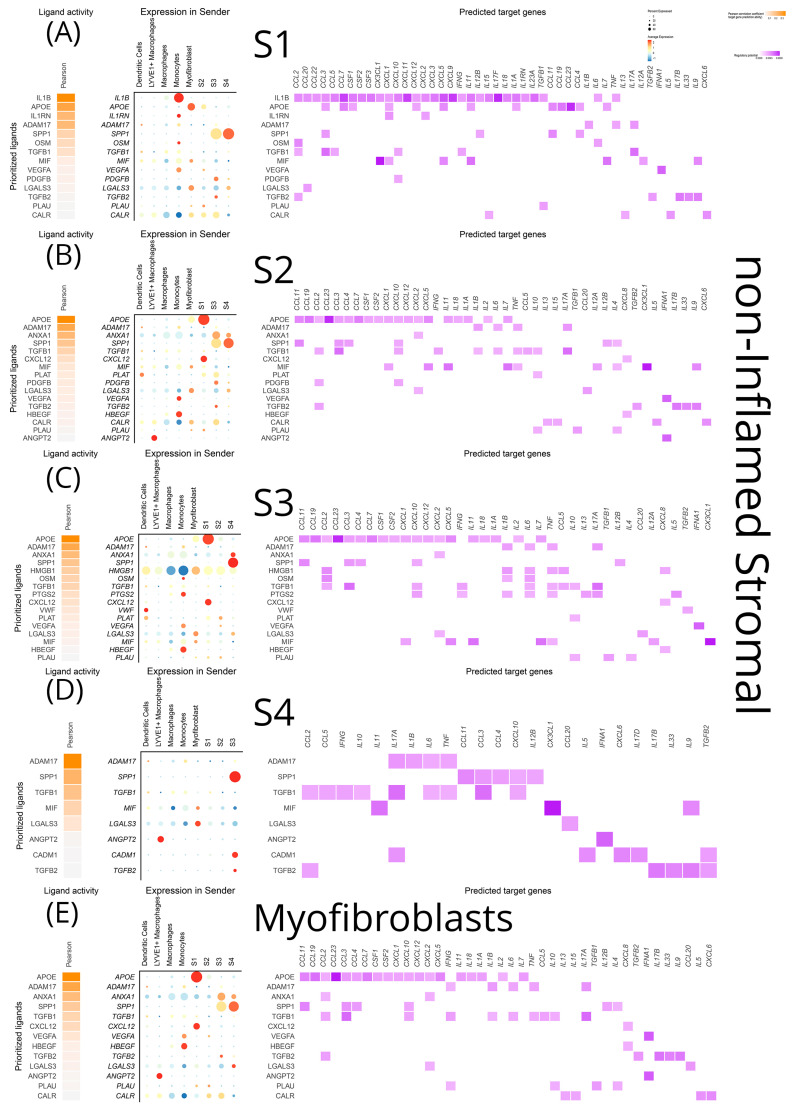
NicheNet visualization of a predictive analysis that establishes connections between cell sources and targets of ligands, as well as effects of the latter on the transcriptional program of the target cells. All stromal clusters were analyzed as both targets and sources of several cytokines, chemokines, growth factors, and signaling molecules within their inflammation status. In each diagram for each stromal cluster, from left to right we depict the top ligands affecting it, their sources, and their effects on the transcriptional program of the target cluster. These plots showcase the signals received by (**A**) non-inflamed S1, (**B**) non-inflamed S2, (**C**) non-inflamed S3, (**D**) non-inflamed S4, (**E**) non-inflamed myofibroblasts.

**Figure 11 biomedicines-12-01674-f011:**
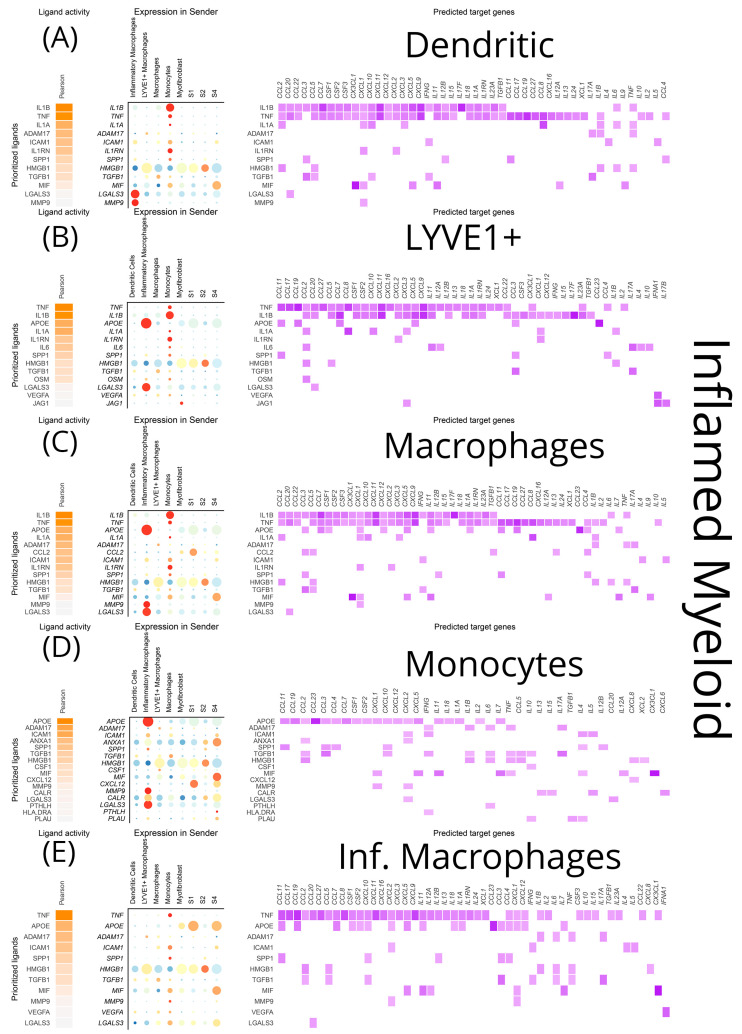
NicheNet visualization of a predictive analysis that establishes connections between cell sources and targets of ligands, as well as effects of the latter on the transcriptional program of the target cells. All myeloid clusters were analyzed as both targets and sources of several cytokines, chemokines, growth factors, and signaling molecules within their inflammation status. In each diagram for each myeloid cluster, from left to right we depict the top ligands affecting it, their sources, and their effects on the transcriptional program of the target cluster. These plots showcase the signals received by (**A**) inflamed dendritic cells, (**B**) inflamed LYVE1^+^ macrophages, (**C**) inflamed macrophages, (**D**) inflamed monocytes, (**E**) inflammatory macrophages.

**Figure 12 biomedicines-12-01674-f012:**
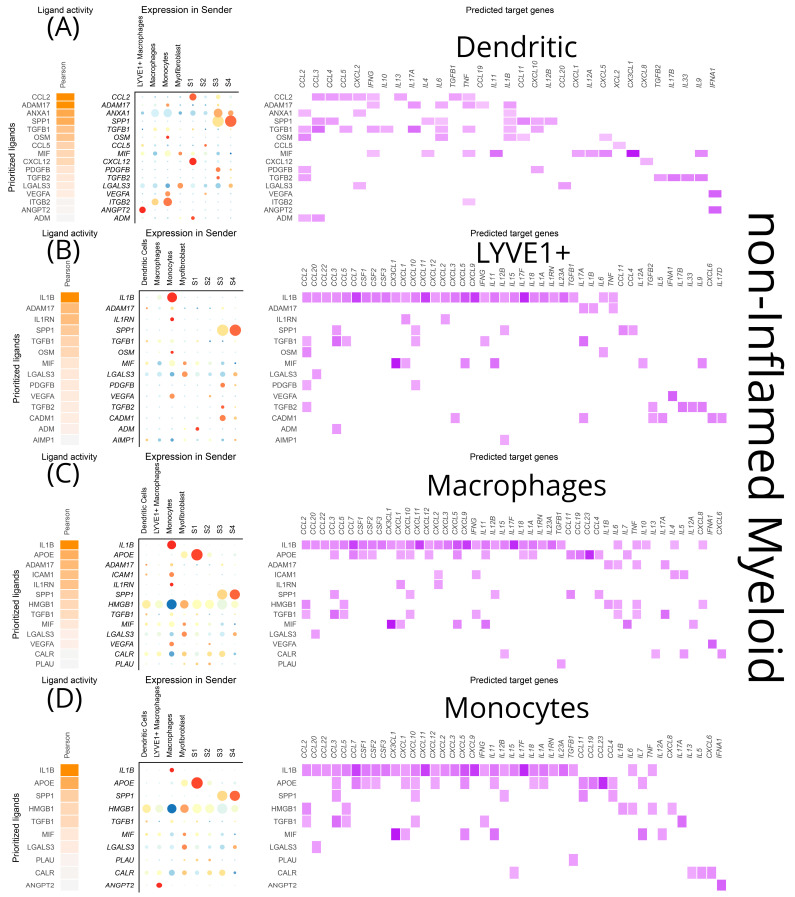
NicheNet visualization of a predictive analysis that establishes connections between cell sources and targets of ligands, as well as effects of the latter on the transcriptional program of the target cells. All myeloid clusters were analyzed as both targets and sources of several cytokines, chemokines, growth factors, and signaling molecules within their inflammation status. In each diagram for each myeloid cluster, from left to right we depict the top ligands affecting it, their sources, and their effects on the transcriptional program of the target cluster. These plots showcase the signals received by (**A**) non-inflamed dendritic cells, (**B**) non-inflamed LYVE1+ macrophages, (**C**) non-inflamed macrophages, (**D**) non-inflamed monocytes.

**Figure 13 biomedicines-12-01674-f013:**
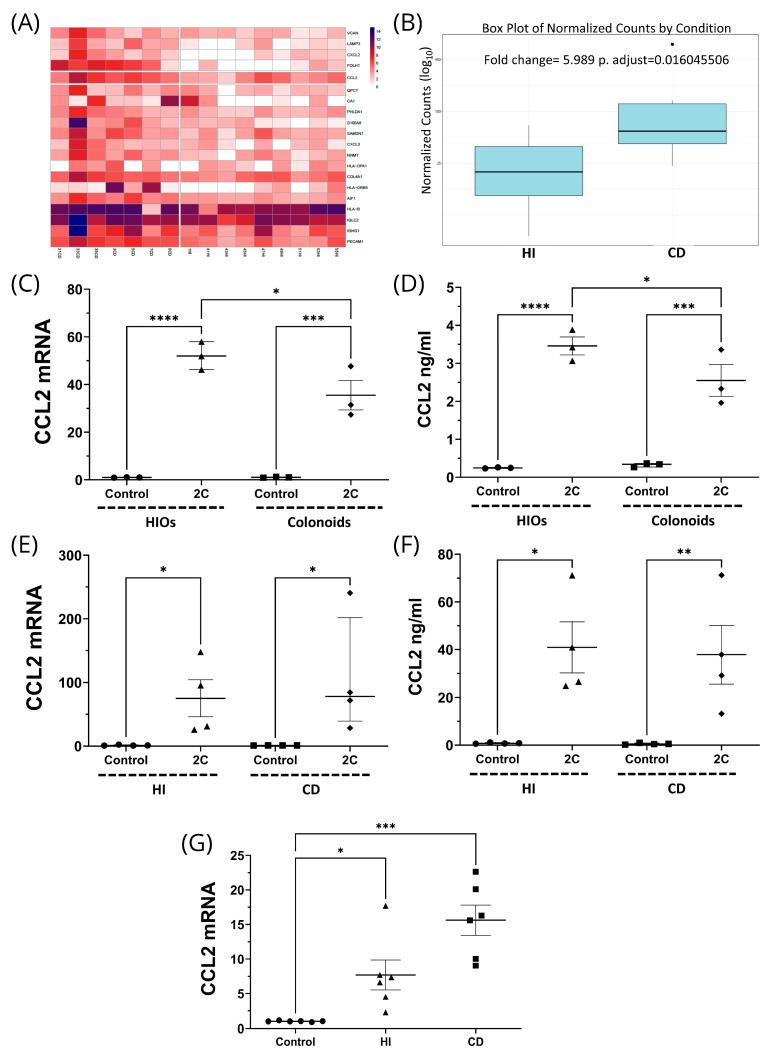
(**A**) Heatmap of RNA sequencing data from mucosal biopsies of 7 Crohn’s disease (CD) patients with ileal involvement (left) and 9 healthy individuals (HI), showing the expression levels of the 20 most differentially expressed genes. The fifth row refers to *CCL2*. (**B**) Boxplots (median, interquartile range) summarizing *CCL2* expression changes per disease status. (**C**) *CCL2* mRNA expression in IL-1α (5 ng/mL)- and TNFα (50 ng/mL)- stimulated (2C) HIOs and colonoids, showing induction in both cases. However, induction was more prominent in HIOs in a statistically significant way. (**D**) Protein secretion pattern of CCL2 in the experimental setting of C was identical. (**E**) Induction of *CCL2* mRNA by 2C in stromal cells from HI and patients with CD. (**F**) Identical CCL2 protein secretion pattern in the experimental setting of E. (**G**) mRNA expression of *CCL2* in HI stromal cells stimulated with biopsy culture supernatants from HI or CD patients, showing that *CCL2* mRNA is upregulated during both cases. ****, *p* < 0.0001; ***, *p* < 0.001; **, *p* < 0.01; *, *p* < 0.05. (**B**) n = 7 for CD, 9 for HI; (**C**,**D**) n= 3; (**E**,**F**) n = 4; G n = 3.

**Table 1 biomedicines-12-01674-t001:** CD patients’ characteristics, from which cSEMFs were isolated.

Patient	Age	Sex	Age of Onset	Disease Location	Disease Behavior	Endoscopic Score (SES-CD)
1	40	M	A2	L3	B2	5
2	46	F	A3	L3	B1	10
3	32	M	A2	L3	B1	8
4	72	M	A3	L1	B1	7

**Table 2 biomedicines-12-01674-t002:** CD patients’ characteristics, from which tissue culture supernatants were collected.

Patient	Age	Sex	Age of Onset	Disease Location	Disease Behavior	Endoscopic Score (SES-CD)
1	52	M	A2	L3	B2	10
2	80	M	A2	L2	B1	8
3	43	M	A3	L2	B1	9

**Table 3 biomedicines-12-01674-t003:** Primer sequences.

Gene	Forward Primer	Reverse Primer	Tm (°C)
CCL2	AGGAAGATCTCAGTGCAGAGG	AGTCTTCGGAGTTTGGGTTTG	60
GAPDH	GACATCAAGAAGGTGGTGAA	TGTCATACCAGGAAATGAGC	60

## Data Availability

All data presented in this study are publicly available online or included in this manuscript: Single-cell RNA-Seq data: Gene Expression Omnibus GSE134809 (https://www.ncbi.nlm.nih.gov/geo/query/acc.cgi?acc=GSE134809, accessed on 24 July 2024); RNA-Seq data: European Nucleotide Archive PRJEB56386 (https://www.ebi.ac.uk/ena/browser/view/PRJEB56386, accessed on 24 July 2024)

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
