# Peer review of "Landscape of Interactions between Stromal and Myeloid Cells in Ileal Crohn’s Disease; Indications of an Important Role for Fibroblast-Derived CCL-2"

_biomedicines, 2024, doi:10.3390/biomedicines12081674_

Round 1

Reviewer 1 Report

Comments and Suggestions for Authors

The manuscript entitled “Landscape of Interactions between Stromal and Myeloid Cells in Ileal Crohn's Disease” by Nicolas Dovrolis et al. was submitted to Biomedicines for possible considerations.

Based on single-cell RNA sequencing data from inflamed and non-inflamed ileal mucosa of Crohn's Disease (CD), the authors conducted the identification of tissue-specific phenotypes and functional specialization of stromal and myeloid cell clusters and changes in phenotypes and functions. They showed that a pathway of CCL2-dependent pathway was relevant to ileal CD, using intestinal stromal cells.

This study indicated that stromal cell-derived CCL2 sustained small-intestinal inflammation in CD and was a valuable therapeutic target. Overall, the study was well designed and written. Some issues should be addressed as follows,

1)      The title was rather big. Yes, the manuscript tried to show the full-frame landscape of interactions between Stromal cells and myeloid cells in Ileal Crohn's disease. But the study was limited to its contents and truly had its focus. I did not notice that everything was described in details in this manuscript. However, CCL2 as the major therapeutic target of CD, should be pointed out in the title. The title should be clear and more specific to CCL2. Giving a wide frame without its focus, does not help the understanding of the manuscript from its title.

2) Table 3, why only a single CCL2 gene was picked up and running RT-qPCR for the confirmation test? How about other genes? What's going on for other targets or cytokines? In general, you should test several candidates together with CCL2, to give a comparison and to confirm the results. Only test CCL2 is just showing the luck, not doing deduced research.

3) Figure 3. Why authors not put the Violin plots vertical instead of the horizontal direction? Each plot was so small to tell.

4) Figure 4, The caption of the figure 4 was not clear.

5) Figure 5 was crowed.

6) Figure 6 was vague to see the details. Better to separate the Figure 6 into two figures.

7) the right and left parts of Figure 7 can be separated into two figures.

8) in case the size of the figures were too difficult to see clearly, full list of the effected genes should be provided as formal Table(s) in the main texts.

9) MIF, IFNγ, and FN1, should also be tested in this study, in parallel to CCL2 molecules.

Major revision should be made on the current manuscript.

Author Response

Comment:
The manuscript entitled “Landscape of Interactions between Stromal and Myeloid Cells in Ileal Crohn's Disease” by Nicolas Dovrolis et al. was submitted to Biomedicines for possible considerations.

Based on single-cell RNA sequencing data from inflamed and non-inflamed ileal mucosa of Crohn's Disease (CD), the authors conducted the identification of tissue-specific phenotypes and functional specialization of stromal and myeloid cell clusters and changes in phenotypes and functions. They showed that a pathway of CCL2-dependent pathway was relevant to ileal CD, using intestinal stromal cells.

This study indicated that stromal cell-derived CCL2 sustained small-intestinal inflammation in CD and was a valuable therapeutic target. Overall, the study was well designed and written. Some issues should be addressed as follows,

Response: 
We thank the reviewer for his/her time and effort in evaluating our manuscript. Please find below point-by-point responses to his/her comments.

Comment:
1) The title was rather big. Yes, the manuscript tried to show the full-frame landscape of interactions between Stromal cells and myeloid cells in Ileal Crohn's disease. But the study was limited to its contents and truly had its focus. I did not notice that everything was described in details in this manuscript. However, CCL2 as the major therapeutic target of CD, should be pointed out in the title. The title should be clear and more specific to CCL2. Giving a wide frame without its focus, does not help the understanding of the manuscript from its title.

Response:
Thank you for this suggestion. This was in principle a bioinformatics approach to analyzing single-cell data with a focus on stromal-myeloid interactions and we believe that the study has achieved its goals. The use of CCL2 as a validation of these bioinformatics insights was put forward because of its well-established role in inflammation but, to our satisfaction, produced some worth-mentioning wet-lab results. We amended the title to: “Landscape of Interactions between Stromal and Myeloid Cells in Ileal Crohn's Disease; Indications of an Important Role for Fibroblast-derived CCL-2”. We hope that it now incorporates both our vision and your suggestion.

Comment:
2) Table 3, why only a single CCL2 gene was picked up and running RT-qPCR for the confirmation test? How about other genes? What's going on for other targets or cytokines? In general, you should test several candidates together with CCL2, to give a comparison and to confirm the results. Only test CCL2 is just showing the luck, not doing deduced research.

Response:
Allow us to reiterate and expand on our previous response. Our work had a bioinformatics focus on single-cell data and CCL2 was only used as a validator of those results, due to its high expression ranking in our RNAseq data. We would be willing to test the proposed genes and even more, but this changes the focus of our study and would require an extended timeframe, beyond the one provided for this revision. We do appreciate the idea proposed by the reviewer but moving on to wet-lab experiments and developing cell cultures and organoids for many of our bioinformatics results, creates a solid basis for future work.

Comments:
3) Figure 3. Why authors not put the Violin plots vertical instead of the horizontal direction? Each plot was so small to tell. 

4) Figure 4, The caption of the figure 4 was not clear.
5) Figure 5 was crowed.
6) Figure 6 was vague to see the details. Better to separate the Figure 6 into two figures
7) the right and left parts of Figure 7 can be separated into two figures.
8) in case the size of the figures were too difficult to see clearly, full list of the effected genes should be provided as formal Table(s) in the main texts

Response:
In our amended manuscript, we split up the aforementioned figures (3-7) into figures 3-12 to improve visibility. All figures were originally submitted in very high resolutions but due to the format of the manuscript provided for reviewing, we understand they were difficult to discern. We also provide these new figures in their full resolutions at: https://figshare.com/s/c329da5634318f82a299 for your consideration.

Comment:
9) MIF, IFNγ, and FN1, should also be tested in this study, in parallel to CCL2 molecules.

Response:
Allow us to refer you to our previous answer to your comment no.2.

Reviewer 2 Report

Comments and Suggestions for Authors

The manuscript addresses an important question from both a basic research and clinical perspective.

The authors worked extensively using advanced techniques to study the molecular and genetic expression of markers of different groups of stromal and myeloid cells and, through computer programs, combined the results to verify the relationship between these cells. They were able to locate these cell clusters in the inflamed or non-inflamed ileal area of ​​people with CD and infer the potential role of specific mediators in the inflammatory process.

Although my review of the text is limited by my partial knowledge of the methodology used, I appreciated the authors' ability to accurately describe complex results.

However, I must intervene by highlighting some theoretical aspects of their way to classify the cells and describe the data that do not meet my point of view and I am asking them to consider my suggestions in the text.

1)      Page 2 lines 66-68. I disagree with the authors' statement. In the intestinal wall, different types of macrophages have in fact been described, some of which have a short half-life and rapid renewal while others have a long life and self-renewal depending on the niches they reside in (see review by Vannucchi, 2022, IJMS) so the authors should be more precise.

2)      2) Page 3 line 74. Writing “stromal fibroblasts” is not accurate since all cells found in connective tissue are stromal cells, in general. Fibroblasts are a type of these cells, easily identifiable in shape and function.

3)      3) For this reason, I must make a general note regarding the classification of stromal cells: I could also accept using a classification that distinguishes the connective cells (which ones? all? And the myofibroblasts? they are also stromal cells) into groups like S1, S2 , S3, S4. It can be useful when investigations are carried out at the molecular level. BUT I DO NOT AGREE TO ADD THE TERM FIBROBLAST TO THIS CLASSIFICATION because, as I said above, fibroblasts are a specific type of stromal cell corresponding to that classified as S1. The others cannot be called fibroblasts, unless they have the same shape, distribution and function as those listed as S1. But they don't have it. In fact, for example and as the authors themselves write, S2 cells are telocytes. So, they are different cells that cannot be called fibroblasts.

4)      The discussion generally consists in highlighting the most significant information obtained from the complex data.

What do the authors means with the expression ‘tissue-specific phenotypes’? All the cells investigated belong to the same tissue, the connective one.

Minor note

The figures are often illegible due to the font being too small.

Author Response

Comment:
The manuscript addresses an important question from both a basic research and clinical perspective.

The authors worked extensively using advanced techniques to study the molecular and genetic expression of markers of different groups of stromal and myeloid cells and, through computer programs, combined the results to verify the relationship between these cells. They were able to locate these cell clusters in the inflamed or non-inflamed ileal area of people with CD and infer the potential role of specific mediators in the inflammatory process.

Although my review of the text is limited by my partial knowledge of the methodology used, I appreciated the authors' ability to accurately describe complex results.

However, I must intervene by highlighting some theoretical aspects of their way to classify the cells and describe the data that do not meet my point of view and I am asking them to consider my suggestions in the text.

Response:
We would like to thank the reviewer for the time and effort in reviewing our manuscript. We appreciate his/her point of view and appreciation of our efforts to explain our work in detail in a way that is directed to a general audience. Please find below point-by-point responses to his/her comments.

Comment:
1)         Page 2 lines 66-68. I disagree with the authors' statement. In the intestinal wall, different types of macrophages have in fact been described, some of which have a short half-life and rapid renewal while others have a long life and self-renewal depending on the niches they reside in (see review by Vannucchi, 2022, IJMS) so the authors should be more precise.

Response:
We understand the reviewer’s point and thank you for the provided direction. We have amended the manuscript to reflect this and now the new lines 64-71 read: Although the intestine becomes seeded during development by embryonic precursors of macrophages, only a fraction of those is maintained to adulthood[8]. A substantial fraction of intestinal macrophages derive from circulating adult monocytes and are constantly replenished during adulthood depending on the niches they reside in [9]. Specifically in the lamina propria (LP) the “physiological inflammation” occurring from the constant exposure to luminal antigens results in a shorted macrophage lifespan. Therefore, a continuous recruitment of monocytes to the intestinal mucosa is necessary during homeostasis that is further intensified in the context of intestinal in-flammation such as IBD[10,11]. 

Comment:
2) Page 3 line 74. Writing “stromal fibroblasts” is not accurate since all cells found in connective tissue are stromal cells, in general. Fibroblasts are a type of these cells, easily identifiable in shape and function.

Response:
We have revised our figures and the whole manuscript to reflect your suggestion.

Comment:
3) For this reason, I must make a general note regarding the classification of stromal cells: I could also accept using a classification that distinguishes the connective cells (which ones? all? And the myofibroblasts? they are also stromal cells) into groups like S1, S2 , S3, S4. It can be useful when investigations are carried out at the molecular level. BUT I DO NOT AGREE TO ADD THE TERM FIBROBLAST TO THIS CLASSIFICATION because, as I said above, fibroblasts are a specific type of stromal cell corresponding to that classified as S1. The others cannot be called fibroblasts, unless they have the same shape, distribution and function as those listed as S1. But they don't have it. In fact, for example and as the authors themselves write, S2 cells are telocytes. So, they are different cells that cannot be called fibroblasts.

Response:
As above, we revised our figures and the whole manuscript to reflect your suggestion..

Comment:
4 ) The discussion generally consists in highlighting the most significant information obtained from the complex data.
What do the authors means with the expression ‘tissue-specific phenotypes’? All the cells investigated belong to the same tissue, the connective one.

Response:
We agree that this phrase did not reflect what we wanted to express. We have amended the conclusions to read: “In conclusion, single-cell RNA sequencing data, from inflamed and non-inflamed ileal mucosa of CD, enabled us to gain insights into the functional specialization of stromal and myeloid cell clusters and revealed changes in phenotype and function from the assumed homeostatic state of the non-inflamed mucosa to that of the inflamed one.”

Comment:
5) The figures are often illegible due to the font being too small

Response:
As per both reviewers’ instructions, we split up the figures 3-7 into 3-12 to improve visibility. All figures were originally submitted in very high resolutions but due to the format of the manuscript provided for reviewing, we understand they were difficult to discern. We also provide these new figures in their full resolutions at: https://figshare.com/s/c329da5634318f82a299 for your consideration.

Round 2

Reviewer 1 Report

Comments and Suggestions for Authors

1) 2. Materials and Methods 

There were no materials to be descripted. All were about methods.

2) for Table2, as you see,  Table 2 was across the two pages. Avoid this issue.

3) As for the new Fig. 4, violin plots were improved a lot, but the whole Figure 4 was still crowded.  

4) Good to see many improvements, but the new Fig. 6 was still rather crowded. Difficult to see each panel of Figure 6 clearly. 

5) authors described the expression of genes that should include both transcriptional and translational levels. Although we saw RT-qPCR data in supporting of the authors conclusions, the protein levels and possible modifications of specific gene should be observed or mentioned, by using western blotting and protein activity, besides ELISA assay in this study. 

6) There are only 4 cited references which were published in 2023, such as ref. 2, ref. 53, ref.65, and ref.75. Kindly recommend authors to update the literatures. Try to include very recently reported relevant researches.

Author Response

We thank the reviewer for his/her continued support in improving our manuscript. Please find below responses to each of your comments.

1)    Comment:
 Materials and Methods. There were no materials to be descripted. All were about methods.

Response:
This wording is based on the template provided by the journal and follows the same format for all articles. 

2)    Comment:
for Table2, as you see,  Table 2 was across the two pages. Avoid this issue.

Response:
Table2 has been amended to fit in the same page as per the reviewer’s comment. 

3)    Comment:
As for the new Fig. 4, violin plots were improved a lot, but the whole Figure 4 was still crowded.  

Response:
Unfortunately, there is no way for us to further divide the figure without omitting important information. We will make sure to work with the editorial team to find solutions to that as necessary.

4)    Comment:
Good to see many improvements, but the new Fig. 6 was still rather crowded. Difficult to see each panel of Figure 6 clearly. 

Response:
This issue lies within the odd number of panels we had to implement in some of our results. We have tried to find the best layout for implementing them. Fortunately, this will be resolved in the online version of the manuscript, which will include zoom functionality for the figures.

5)    Comment:
authors described the expression of genes that should include both transcriptional and translational levels. Although we saw RT-qPCR data in supporting of the authors conclusions, the protein levels and possible modifications of specific gene should be observed or mentioned, by using western blotting and protein activity, besides ELISA assay in this study. 

Response:
We thank the reviewers for his/her insights into the manuscript. The wet lab data we present at the levels of transcription and protein level serve their purpose as an indication that the in silico data we present are in alignment with in vitro and ex vivo findings. Performing western blot to confirm our ELISA results and chemotaxis (activity) assays with stromal and monocyte cell co-cultures to further study specific monocyte-stromal cell interactions is a mandatory starting point of a new, focused specifically on chemokines, project that we intend to pursue in the future.

6)    Comment:
There are only 4 cited references which were published in 2023, such as ref. 2, ref. 53, ref.65, and ref.75. Kindly recommend authors to update the literatures. Try to include very recently reported relevant researches.

Response:
Per the reviewer’s instructions, we have further enriched our manuscripts with newer studies

Reviewer 2 Report

Comments and Suggestions for Authors

I thank the authors for their commitment to following my suggestions.

Greater clarity has been made in the naming of the numerous and different connective cells.

Author Response

We thank the reviewer again for his/her important feedback